

Atmospheric
Measurement
Techniques

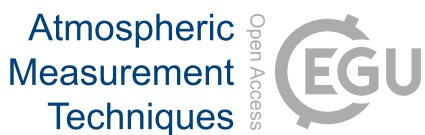

# Evaluation of aerosol microphysical, optical and radiative properties measured with a multiwavelength photometer

**Yu Zheng[1], Huizheng Che[1], Yupeng Wang[2], Xiangao Xia[3], Xiuqing Hu[4], Xiaochun Zhang[5], Jun Zhu[6], Jibiao Zhu[1], Hujia Zhao[7], Lei Li[1], Ke Gui[1], and Xiaoye Zhang[1]**

[1]State Key Laboratory of Severe Weather & Key Laboratory of Atmospheric Chemistry,
Chinese Academy of Meteorological Sciences, China Meteorological Administration, Beijing, 100081, China
[2]Changchun Institute of Optics, Fine Mechanics and Physics, Chinese Academy of Sciences, Changchun, 130033, China
[3]LAGEO, Institute of Atmospheric Physics, Chinese Academy of Sciences, Beijing, 100029, China
[4]National Satellite Meteorological Center, China Meteorological Administration, Beijing, 100081, China
[5]Centre for Atmosphere Watch and Services, Meteorological Observation Centre,
China Meteorological Administration, Beijing, 100081, China
[6]Jiangsu Key Laboratory of Atmospheric Environment Monitoring and Pollution Control,
Collaborative Innovation Center of Atmospheric Environment and Equipment Technology,
School of Environmental Science and Engineering, Nanjing University of Information Science & Technology,
Nanjing, 210044, China
[7]Institute of Atmospheric Environment, China Meteorological Administration, Shenyang 110166, China

**Correspondence:** Huizheng Che (chehz@cma.gov.cn)

**Abstract.** An evaluation of aerosol microphysical, optical and radiative properties measured with a multiwavelength photometer named CW193 was performed in this study. The instrument has a highly integrated design, smart control performance and is composed of three parts (the optical head, robotic drive platform and stents system). Based on synchronous measurements, the CW193 products were validated using reference data from the AERONET CE318 photometer. The results show that the raw digital counts from CW193 agree well with the counts from AERONET ($R > 0.989$), with daily average triplets of around 1.2 % to 3.0 % for the ultraviolet band and less than 2.0 % for the visible and infrared bands. Good aerosol optical depth agreement ($R > 0.997$, 100 % within expected error) and root mean square error (RMSE) values ranging from 0.006 (for the 870 nm band) to 0.016 (for the 440 nm band) were obtained, with the relative mean bias (RMB) ranging from 0.922 to 1.112 and the aerosol optical depth bias within $\pm 0.04$. The maximum deviation of the peak value for fine-mode particles varied from about 8.9 % to 77.6 %, whereas the variation for coarse-mode particles was about 13.1 % to 29.1 %. The deviation variations of the single scattering albedo were approximately 0.1 %–1.8 %, 0.6 %–1.9 %, 0.1 %–2.6 % and 0.8 %–3.5 % for the 440, 675, 870 and 1020 nm bands, respectively. For the aerosol direct radiative forcing, deviations of approximately 4.8 %–12.3 % were obtained at the earth's surface and 5.4 %–15.9 % for the top of the atmosphere. In addition, the water vapor retrievals showed satisfactory accuracy, characterized by a high $R$ value ($\sim 0.997$), a small RMSE ($\sim 0.020$) and a good expected error distribution (100 % within expected error). The water vapor RMB was about 0.979, and the biases mostly varied within $\pm 0.04$, whereas the mean values were concentrated within $\pm 0.02$.

## 1 Introduction

Atmospheric aerosols have a substantial impact on the whole environment, and affect the regional air quality and global climate change in particular. In terms of the earth's climate, aerosols represent one of the determining factors for climate change for which there are also large uncertainties (Che et

al., 2019a; Gui et al., 2017; Hansen et al., 1997; Letu et al., 2020b; Xing et al., 2020; Zhao et al., 2021a). Specifically, atmospheric aerosols can disturb the earth's radiative budget and modify it not only by scattering and/or absorbing the in-cident solar radiation and the outgoing radiation from the sur-face (aerosol direct radiative effects) but also by altering the microphysical properties of clouds, such as the cloud con-densation nuclei concentration and reflectivity (Charlson et al., 1992; Dubovik et al., 2002; Letu et al., 2020a; Zhao et al., 2020). In addition, the distribution of aerosols in the at-mosphere is not uniform, and is characterized by high spatial and temporal variability among regions (Gui et al., 2021a; Li et al., 2020a; Zhao et al., 2021b). For these reasons, an inte-grated and accurate understanding of aerosol microphysical, optical and radiative properties is essential for studies on the climatic and environmental effects of aerosols, particularly when assessing the response of the climate to anthropogenic aerosols (Bi et al., 2014; Che et al., 2019c; Holben et al., 1998; Miao et al., 2021). At present, the two main techniques used to monitor variations in columnar aerosol optical prop-erties are remote sensing by satellites and ground-based ob-servations. As revealed by previous studies, the aerosol opti-cal depth (AOD) and Ångström exponent are the most com-monly used and important parameters for investigating the features of aerosols, and are widely used in numerical mod-eling and satellite calibration (Li et al., 2020b; Zhang et al., 2021a, b; Zhao et al., 2018).

Remote sensing from satellite-borne platforms has devel-oped rapidly since its inception, owing to its powerful fea-tures and convenience, especially for the global and long-term observation of atmospheric aerosols (Gui et al., 2019, 2021c; Ma et al., 2021). For example, the Advanced Very High Resolution Radiometer (AVHRR) (Hauser et al., 2005; Stowe et al., 1997) and the Total Ozone Mapping Spec-trometer (TOMS) (Hsu et al., 1999) have provided long-term global AOD products from 1979 to the present. The Sea-viewing Wide Field-of-view Sensor (SeaWiFS) and the Visible Infrared Imaging Radiometer Suite (VIIRS) provide aerosol retrieval products such as the fine-mode fraction and the particle densities of aerosols (Gordon and Wang, 1994; Sayer et al., 2012). In recent years, a series of ad-vanced satellite sensors for aerosol monitoring have been launched, such as the Multi-angle Imaging SpectroRadiome-ter (MISR) (Garay et al., 2017), the Moderate Resolution Imaging Spectrometer (MODIS) (Wei et al., 2019), and Cloud-Aerosol Lidar and Infrared Pathfinder Satellite Ob-servations (CALIPSO) (Kim et al., 2018), which have con-tributed greatly to our understanding of the temporal and spa-tial distribution characteristics of aerosols. Nevertheless, as Li et al. (2020a) reported, there are still considerable uncer-tainties in the satellite AOD retrieval products due to the in-fluences of sensor calibration, cloud contamination and sur-face albedo. In addition, owing to the limitation on the tem-poral resolution of satellite-borne platforms over specific re-gions such as high-altitude areas and huge-emission areas,

satellite AOD retrieval products cannot meet the advanced requirements for ecological environment assessment, heath effect studies and real-time monitoring (Dubovik et al., 2006; Gui et al., 2021b; Ma et al., 2021; Miao et al., 2019; Ningom-bam et al., 2021; Xing et al., 2021b; Zheng et al., 2019).

For these reasons, aerosol detection from ground-based observations is regarded as the most direct, accurate and ef-fective technique to measure and study the columnar mi-crophysical, optical and radiative properties of atmospheric aerosols, and there are extensive ground-based monitor-ing networks across the world dedicated to aerosol detec-tion, such as the Precision Filter Radiometer (PFR) net-work of the Global Atmosphere Watch program of the World Meteorological Organization (WMO-GAW; Cuevas et al., 2019), the China Aerosol Remote Sensing NETwork (CARSNET; Che et al., 2015, 2018), the Aerosol Robotic Network (AERONET; Holben et al., 1998), the PHOtométrie pour le Traitement Opérationnel de Normalisation Satelli-taire (PHOTONS; Goloub et al., 2008) and the SKYrad Net-work (SKYNET; Nakajima et al., 2020), all consisting of precise instruments with the 0.02 AOD accuracy suggested by the WMO (Che et al., 2009). Most of these observation networks are equipped with the CE318 (Cimel Electronique, France), an automatic multiband sun photometer that is used as the master and/or observation instrument to provide long-term data on aerosol microphysical, optical, and radiative characteristics at the global scale. These networks play an important role in determining the climatic and environmen-tal effects of aerosols, especially in polar and plateau regions where robotic measurements may be a better choice due to the harsh climatic conditions and lack of manpower sup-port, and their measurement results have been strictly veri-fied under a wide range of conditions (Dubovik et al., 2000; Eck et al., 1999; Xing et al., 2021a; Zhuang et al., 2017). As discussed in WMO-GAW report nos. 162, 207, 227 and 228 (from 2004, 2012, 2016 and 2017, respectively), the multiwavelength aerosol optical depth (AOD) is still recom-mended as the long-term measurement variable in the im-plementation plan from 2016 to 2023. Ground-based AOD attenuation observation in particular is regarded as a highly accurate monitoring method that provides indispensable data for satellite validation and global modeling. According to this guideline, the absolute limit on the estimated uncertainty is 0.02 optical depths for acceptable data, and < 0.01 is the goal to be achieved in the near future. Additionally, the in-ternational coordination of AOD networks is still inadequate and could be improved by a federated network under the WMO-GAW umbrella, and networks should be made trace-able and maintainable via intercomparisons and calibrations. These guidelines highlight that data assessment is as impor-tant as field observations. However, in China, due to the vast territory and various landforms present, there are still many observation gaps in the monitoring of aerosol optical prop-erties. Furthermore, the complicated underlying surface and emission conditions result in distinct temporal and spatial

variations in the aerosol. Therefore, referring to the WMO-GAW's criterion, conducting field observations and evaluating data are of great importance for reducing the uncertainties in aerosol optical properties, which will be of great assistance when combating climate change.

Many photometers aside from CE318, POM-02 (Nakajima et al., 2020) and PFR (Kazadzis et al., 2018) have been used for AOD measurements in China, such as DTF-5 and PSR-2 (Li et al., 2012; Huang et al., 2019). However, we suggest that all such instruments and their products should meet the WMO-GAW's criterion and remain consistent with AERONET, thus providing comprehensive, comparable aerosol optical products. Here, we present a highly integrated multiwavelength photometer named CW193 (CW is an abbreviation of "Chinese device for World") for monitoring aerosol microphysical, optical and radiative properties. It has a user-friendly instruction system and most of its components are assembled in a robotic drive platform, which makes the whole system efficient and highly integrated. By using direct sun and diffuse-sky radiation measurements, the CW193 not only provides the columnar instantaneous AOD but also gives detailed information on the aerosol characteristics, including – but not limited to – the volume size distribution (VSD), the single scattering albedo (SSA), the asymmetry factor (ASY) and the aerosol direct radiative forcing (ADRF), which are the key input parameters for numerical models (Miao et al., 2020; Stier et al., 2005; Wang et al., 2013). These features make the CW193 a particularly suitable multiwavelength photometer for monitoring aerosol microphysical, optical and radiative properties, which can contribute to verifying satellite and modeling products.

For this study, synchronous measurements were conducted by CW193 and CE318s from AERONET and CARSNET at CAMS (Chinese Academy of Meteorological Sciences), and the products of CW193 were evaluated and compared in detail with the reference of AERONET to check that they remain consistent with it. Following the present introduction, the observation site and ancillary information for this study are introduced in Sect. 2. In Sect. 3, a description of the new CW193 multiwavelength photometer is provided. Section 4 provides an evaluation and comparison of the aerosol microphysical, optical and radiative properties from CW193. Finally, the main conclusions are presented in Sect. 5.

## 2    Observation site and ancillary information

### 2.1    Observation site

In this study, the CW193 instrument was tested in the atmospheric composition observation platform of CAMS (116.317° E, 39.933° N, 106 m a.s.l., see Fig. 1), in the north urban area of Beijing, where the main forms of pollution are derived from urban activities. As revealed by Che et al. (2015, 2019b) and Zheng et al. (2019), according to long-

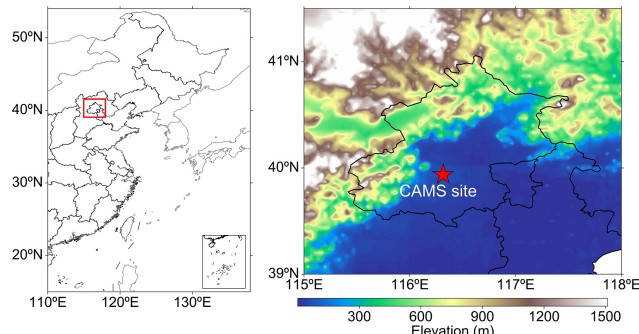

**Figure 1.** Location of the CAMS site.

term ground-based aerosol measurements at CAMS, the annual mean $AOD_{440\,nm}$ is $\sim 0.65 \pm 0.60$, with a maximum monthly mean of $\sim 0.82 \pm 0.77$ in July and a minimum monthly mean of $\sim 0.39 \pm 0.41$ in December, which are considered to be representative of the urban atmospheric conditions in China and a good test environment for CW193. This CAMS site (named "Beijing-CAMS") is part of the AERONET observation network and has provided the AOD and other inversion products since its establishment in 2012. In addition, Beijing-CAMS is a transfer sun calibration site for CARSNET, with the master instruments sent to the Izaña Observatory (Izaña, Canary Islands, Spain; 28.3° N, 16.5° W, 2373 m a.s.l.) for annual calibration. The data on particulate matter (PM) concentrations used in this study were provided by the Ministry of Ecology and Environment of the People's Republic of China (https://air.cnemc.cn:18007/, last access: 29 March 2022).

### 2.2    Ancillary information

#### 2.2.1    CE318 sun photometer and its observation network

In this comparative observation campaign, the AOD data and their correlative aerosol inversions provided by AERONET and CARSNET were used to validate the results retrieved from the CW193 observations. AERONET is the biggest federated instrument network in the world, providing open-access data on aerosol microphysical, optical and radiative properties (https://aeronet.gsfc.nasa.gov/, last access: 29 March 2022). CARSNET is the largest ground-based aerosol remote-sensing network in China, with more than 80 sites in China, of which 51 are currently operational. CARSNET uses a similar algorithm to AERONET (Dubovik et al., 2000; Dubovik and King, 2000) and has a rigorous calibration process; therefore, the aerosol retrievals of CARSNET are of great importance for determining the temporal and spatial variations of aerosol optical properties in China (Che et al., 2018; Yu et al., 2015; Zhao et al., 2021b; Zheng et al., 2021).

The master instrument used in AERONET and CARSNET is the CE318 sun photometer, which performs direct sun and

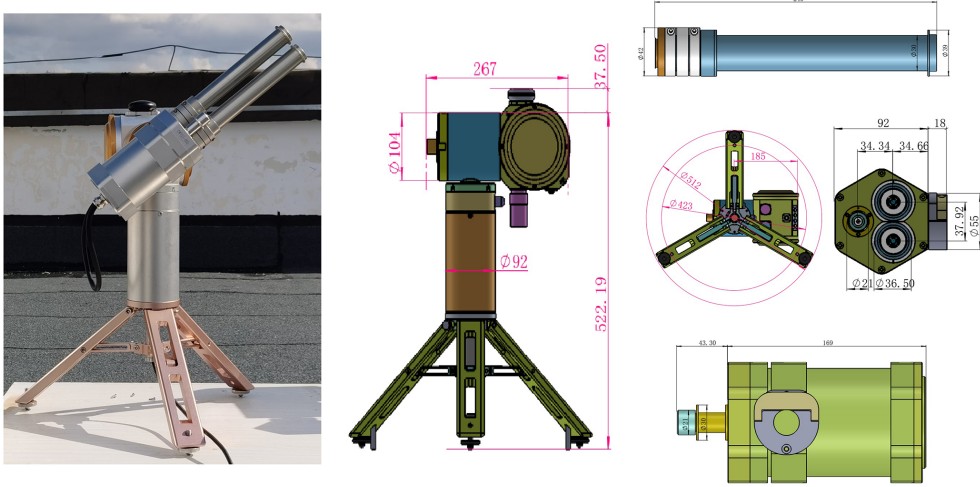

**Figure 2.** CW193 scheme and dimensions.

diffuse-sky radiation measurements at set obs[...] [...]ecisely coated For the direct sun measurements, the radiati[...] [...]ce. There are at 340, 380, 440, 500, 675, 870, 1020 and 164[...] [...]idity and temperature-late an accurate AOD, and at 936 nm for wate[...] [...]nformation is with uncertainties within ±0.02 and ±0.10 cm[...] [...]he raw signal, The diffuse-sky measurements are conducted at 440, 500, 670, 870, 1020 and 1640 nm to retrieve the microphysical and optical properties of aerosols in different routines: the almucantar (ALM) and the principal plane (PPL). The azimuth angle is varied while the zenith angle is kept constant for the ALM, and vice versa for the PPL. In this study, the CE318s and CW193 were set to perform intensive direct sun observations every 3 min (normally, they would be performed every 15 min) to obtain enough data to evaluate the AOD accuracy.

### 2.2.2 CW193 multiwavelength photometer

The CW193 is an automatic photometer that is designed to obtain AOD and other retrievals (such as microphysical, optical and radiative properties of aerosols) from solar radiation and sky radiation monitoring. The instrument is mainly composed of three parts: the optical head, robotic drive platform and stents system (as shown in the left part of Fig. 2). These three parts can be easily connected together using only a few screws. Aside from its highly integrated design, the cross weight of CW193 is about 12 kg, which makes it easier to transport. We present a comparison of the technical specifications of CE318-T and CW193 in Table 1.

Two collimators with a 1.30° full field of view are both separately screwed tightly to the optical head (making its disassembly and maintenance more convenient) to avoid interference from stray light and to reduce the measurement error originating from nonparallel integrated collimators, as used in CE318. To compare the results with AERONET, the detector in the optical head is designed with nine optical filters with nominal wavelengths centered at 340, 380, 440, 500,

thus minimizing the temperature dependence of the silicon detectors at 1020 and 1640 nm.

The robotic drive platform is the main dynamic system that allows the optical head to track the direct solar radiation, and is used in the ALM scan routines. To avoid mechanical problems owing to excessive usage of the robotic platform, CW193 is designed to keep tracking the sun all the time unless the ALM routines are activated at specific integral local times (09:00, 10:00, 11:00, 12:00, ...). In addition, all the measurement routines are suspended when precipitation is detected by the wet sensor of the platform, and the optical head will then turn down to avoid rain contamination. On the whole, the system is protected up to the IP65 level, making it tough enough to run in a humid or dusty environment.

The stents system, which is directly supported on the base of the robotic drive platform, consists of an adjustable-length tripod with a horizontal adjustment knob at each foot; therefore, it can be quickly deployed and fixed onto flat and/or rigid surfaces and has a reliable anti-wind capacity ($< 25\,\mathrm{m\,s^{-1}}$ if not fixed on the ground). The instrument is powered by a 220 V alternating current and is equipped with a solar panel for use in remote locations and in temporary/movable observation campaigns. As a result, the design of CW193 is very robust, ensuring long-term, steady operation at a wide range of temperatures and humidities: between about −30 and 60 °C and between about 0 % and 100 %, respectively.

The main circuit board is in the head of the robotic drive platform and integrates operation control, data acquisition, data storage, transmission communication and status diag-

**Table 1.** Technical specifications for CE318-T[a] TS3 and CW193.

| | CE318-T[a] TS4 | CW193 |
|---|---|---|
| Main components | Optical head, control unit, robot, | Optical head, robotic drive platform, stents system |
| Spectral range | 340, 380, 440, 500, 675, 870, 937, 1020, 1640 nm | 340, 380, 440, 500, 675, 870, 937, 1020, 1640 nm |
| Field of view | 1.26° | 1.30° |
| Azimuth range for detection | 0 to 360° | 0 to 360° |
| Zenith range for detection | 0 to 180° | 0 to 180° |
| Sun tracking accuracy | 0.01° | 0.02° |
| Communication outputs | RS232, USB, UMTS/3G/W-CDMA, GPRS | RS232, USB, 4G |
| Storage | Flash memory (4 MB), SD card (32 G) | Flash memory (32 GB) |
| Power demand | DC 12 V or solar panel (5 W) and external batteries (12 V, 16 Ah) | DC 12 V |
| Software | PhotoGetData | DataMonitor |

[a] CE2 Parameters for the standard version of the CE318-T TS5 photometer.

nosis. The control unit is designed to conduct observations automatically in the default state once the geographic information for the observation site has been confirmed by the built-in BDS (BeiDou Navigation Satellite System) module. The data unit comprises an internal data logger and 32 GB of memory, which is considered lifetime storage with a daily data size of $\sim 150$ kB. Data transmission to a computer can be realized in two ways: serial communication via RS-232 or via the 4G network. The diagnostic module checks the whole system when the instrument is powered on, and the running state is easily recognized by checking the indicator light in the optical head.

The system provides a friendly user interface on a computer, which makes CW193 easy to operate, convenient to maintain and highly functional. In Fig. 3, the functional area and monitoring area are clearly presented in the left and right parts of the interface, respectively. It is very convenient to receive data via the 4G network when serial communication is unavailable (in some remote regions). Also, multiple device control is achievable (devices 003, 005 and 006 are online and controllable in Fig. 3) in this mode CE3. In the data download part, the history data can easily be downloaded by selecting the start and end times via a drop-down menu. All observation instructions are provided in the control commands area and can be sent to the device in the dialog box. In the monitoring area in the right half, the plot and the data plotted are located in the top and bottom windows, respectively, making it convenient for monitoring the device's status. We present a comparison of the functional specifications of CE318-T TS6 and CW193 in Table 2.

### 2.2.3 Calibration and data processing

In this work, the direct sun calibration of CW193 was conducted at the atmospheric composition observation platform of CAMS (one of the calibration centers of CARSNET) using the method of coefficient transfer (intercomparison) with the reference master instruments of AERONET (Che et al.,

2009, 2019c; Zheng et al., 2021). The sphere calibration was performed at the optical calibration laboratory (CAMS, Beijing) of CARSNET by integrating the sphere. We conducted 50 measurements of the sphere's radiance and found extremely small fluctuations in the CW193 digital counts ($< 1$‰), indicating excellent detection stability and accurate sphere calibration coefficients (Tao et al., 2014).

Dear editor and typesetter,

This is my typing error, as the CE318's diffuse-sky observation are conducted at 440, 500, 675, 870, 1020 and 1640 nm wavelength. Also, it can be seen at line 3 in this page, the available wavelengths of CE318 are 340, 380, 440, 500, 675, 870, 936, 1020 and 1640 nm. So, we hope to change this "670" to "675" here.

Thanks a lot.

2021c, Zheng et al., 2021). A verification of the algorithm is provided in the Supplement, guaranteeing the accuracy of this campaign (Figs. S1 and S2 in the Supplement). As for the inversions of VSD and SSA in this campaign, they were retrieved from observational data obtained from diffuse-sky measurements by CW193 at 440, 670 TS7, 870 and 1020 nm using the algorithms of Dubovik et al. (2002, 2006). The ADRF was calculated by the radiative transfer module, which is similar to the inversion of AERONET (García et al., 2008, 2012). Because these inversions and their algorithms have been introduced, validated and applied in many previous studies based on CARSNET observations, we will not do so again in this paper (Che et al., 2018, 2019c; Zhao et al., 2018; Zheng et al., 2021). In general, the uncertainty in the AOD was 0.01–0.02 (Eck et al., 1999). The VSD accuracy was 15 % to 25 % for $0.1\,\mu m \leq r \leq 7.0\,\mu m$ and 25 % to 100 % for other radii (Dubovik et al., 2002). The SSA accuracy was 0.03 when it was calculated under the condition $AOD_{440\,nm} > 0.50$ with a solar zenith angle of $> 50°$ (Dubovik et al., 2002). The bias for measured radiation at the surface was about $9 \pm 12$ W m$^{-2}$, and was affected by the dominant aerosol type (García et al., 2008).

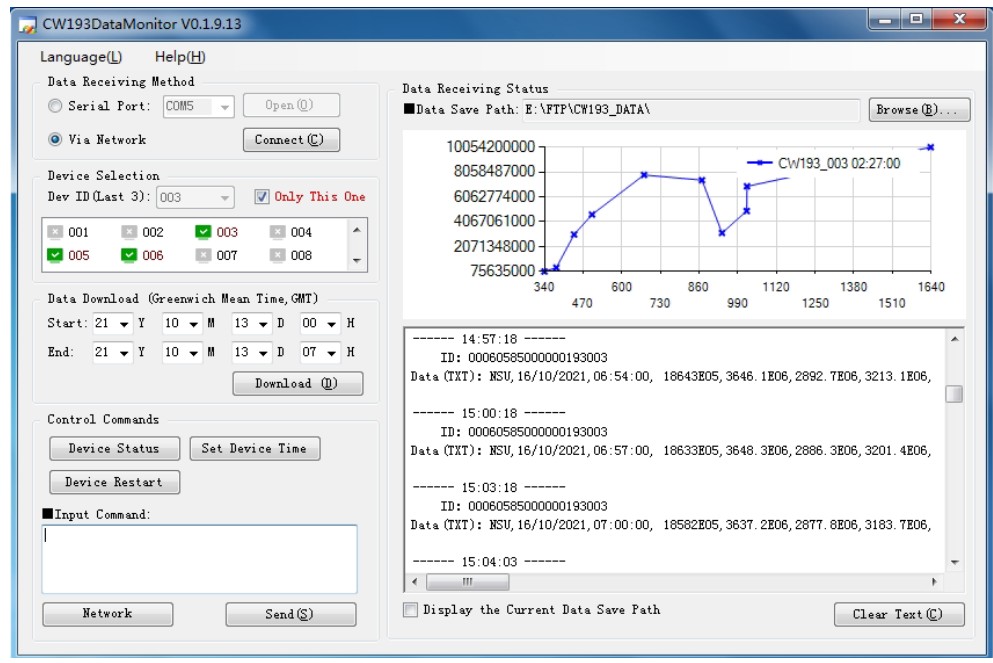

**Figure 3.** Monitoring software of CW193.

## 3   Results and discussion

In this work, synchronous measurements with five instruments were conducted at the CAMS observation platform during 1–11 November 2020. Specifically, photometers #543 and #746 of the CE318-N model and photometers #1043 and #1046 of the CE318-T model are the four master instruments at the Beijing-CAMS site in AERONET, and the raw data from them are transmitted in real time to AERONET. The AODs and other inversion products from these four instruments can be downloaded from the AERONET website. Furthermore, these four instruments are the reference instruments of CARSNET, and play an important role in the operational observations and annual calibration of CARSNET.

### 3.1   Raw digital count evaluation

The raw digital counts are the deciding factor in the precision of the calculation and retrieval results, which reflects the running status and stability of the instrument. In Table 3, we show the observed signal with the least squares method, presenting a basic statistical intercomparison at coincident spectral wavelengths. It is notable that these instruments perform three measurements within $\sim 30$ s in one scenario, and we calculated the average digital count for each observation in this comparison. Furthermore, the results from the AERONET webpage during this campaign were mainly derived from photometers #1043 and #1046 according to the "instrument number" in the downloaded files; therefore, we used the corresponding observation signals of these two mas-

ter instruments to carry out the performance evaluation of CW193. In addition, to avoid the effect of instantaneous atmospheric disturbance, only the values for which the difference in observation time compared to the master instruments was within 20 s were selected and considered effective data in this work.

[Dear copy-editor, typesetter and editor,

We found a type error here, and would like to change this "0.9985" to "0.9995".
Here we intended to showed the $R^2$ values for #1043 and #1046 are higher than 0.9971 and 0.9995 from 340 to 870 nm bands. Please find the Table.2 as below, there is no "0.9985" in the column of $R^2$ for #1046. Because the "8" key is close to the "9" key at the keyboard, i made this type error here and would like to get the approval to revise it. And it should be "0.9995" here.

Thank you very much!]

In practical terms, CW193 performs better in the ultraviolet (UV) bands (340 nm and 380 nm) and visible bands (440 to 870 nm), yielding $R^2$ values larger than 0.9971 and 0.9985 with photometers #1043 and #1046, respectively. The $R^2$ values were relatively low in the infrared bands of 1020 and 1640 nm. The minimum $R^2$ values were $\sim 0.9792$ with photometer #1043 at 1020 nm and $\sim 0.9988$ with photometer #1046 at 1640 nm, indicating greater variation in these two bands than in the other bands owing to their temperature sensitivity (Che et al., 2011; Tao et al., 2014). For the WV channel at 936 nm, the $R^2$ values were $\sim 0.9977$ and $\sim 0.9997$ for photometers #1043 and #1046, respectively;

**Atmos. Meas. Tech., 15, 1–20, 2022**                                    **https://doi.org/10.5194/amt-15-1-2022**

**Table 2.** Functional specifications for CE318-T[a] TS9 and CW193.

| | CE318-T TS10 | CW193 |
|---|---|---|
| Observation frequency for sun measurement | 15 min (default), up to 2 min | 3 min (default), up to 2 min |
| Mode of sun tracking | At the beginning of every measurement | Keep tracking with continuous rotation |
| Observation frequency for ALM scan | According to air mass, when air mass $= 1.7, 2.0, 2.2, 2.4, 2.6...$ | Every integral local time at 07:00, 08:00, 09:00, 10:00, 11:00, ..., 19 o'clock (primary); according to air mass, when air mass $= 1.7, 2.0, 2.2, 2.4, 2.6, ...$ (subsidiary) |
| Observation schedule[b] | Sun, moon, black, principal plane, almucantar, hybrid, cross sun, cross moon. Curvature cross | – Sun, black, almucantar, principal plane (default)<br>– Only sun (optional, consecutive)<br>– Only almucantar (optional, consecutive)<br>– Only principal plane (optional, consecutive) |
| Monitoring software | – Instrument setup[b]<br>– Wavelength selection<br>– Scan mode and scenario configuration<br>– Measurement scheduling<br>– Data analysis<br>– Data visualization<br>– Data storage (raw data, k8, ASCII files) | – Scan mode and scenario configuration<br>– Measurement scheduling<br>– Wavelength selection<br>– Data visualization<br>– Data retrieval<br>– Data storage (TXT files)<br>– Command inputs<br>– Multidevice control (4G mode only) |

[a] Standard version of the CE318-T photometer. CE5 [b] For the photometer in auto mode.

**Table 3.** Coefficient of determination ($R^2$) and number of coincident data ($N$) for the raw digital count comparison between CW193 and CE318 measurements for the nine spectral bands used in this study.

| Wavelength | #1043 | | | #1046 | | |
|---|---|---|---|---|---|---|
| (nm) | $R$ | $R^2$ | $N$ | $R$ | $R^2$ | $N$ |
| 340 | 0.9997 | 0.9994 | 162 | 0.9998 | 0.9996 | 355 |
| 380 | 0.9995 | 0.9990 | 162 | 0.9998 | 0.9996 | 355 |
| 440 | 0.9997 | 0.9994 | 162 | 0.9998 | 0.9997 | 355 |
| 500 | 0.9991 | 0.9982 | 162 | 0.9997 | 0.9995 | 355 |
| 675 | 0.9993 | 0.9985 | 162 | 0.9999 | 0.9998 | 355 |
| 870 | 0.9985 | 0.9971 | 162 | 0.9999 | 0.9998 | 355 |
| 936 | 0.9989 | 0.9977 | 162 | 0.9998 | 0.9997 | 355 |
| 1020 | 0.9895 | 0.9792 | 162 | 0.9995 | 0.9990 | 355 |
| 1640 | 0.9923 | 0.9846 | 162 | 0.9994 | 0.9988 | 355 |

hence, CW193 showed good detection ability for columnar WV.

The triplet value, defined as (maximum − minimum)/mean $\times$ 100 %, is a more effective parameter for obtaining a better description of the stability of the instrument and the atmospheric conditions. Thus, we calculated the triplet for each band, and the diurnal variation in these triplets is shown in Fig. 4. In this study, it can be clearly seen that the triplets showed a typical diurnal distribution, as found in many previous studies (Barreto et al., 2016; Che et al., 2011; Estellés et al., 2012), which is characterized by increasing dispersion with increasing airmass. However, cloud contamination is also an important factor affecting triplet variation. Using the weather record and the cloud-screening results of AERONET (version 3.0) as a reference, we found that the atmospheric conditions on 7 and 11 November were greatly influenced by cloud processes. As a result, the dispersion of the triplets on those 2 days was larger than

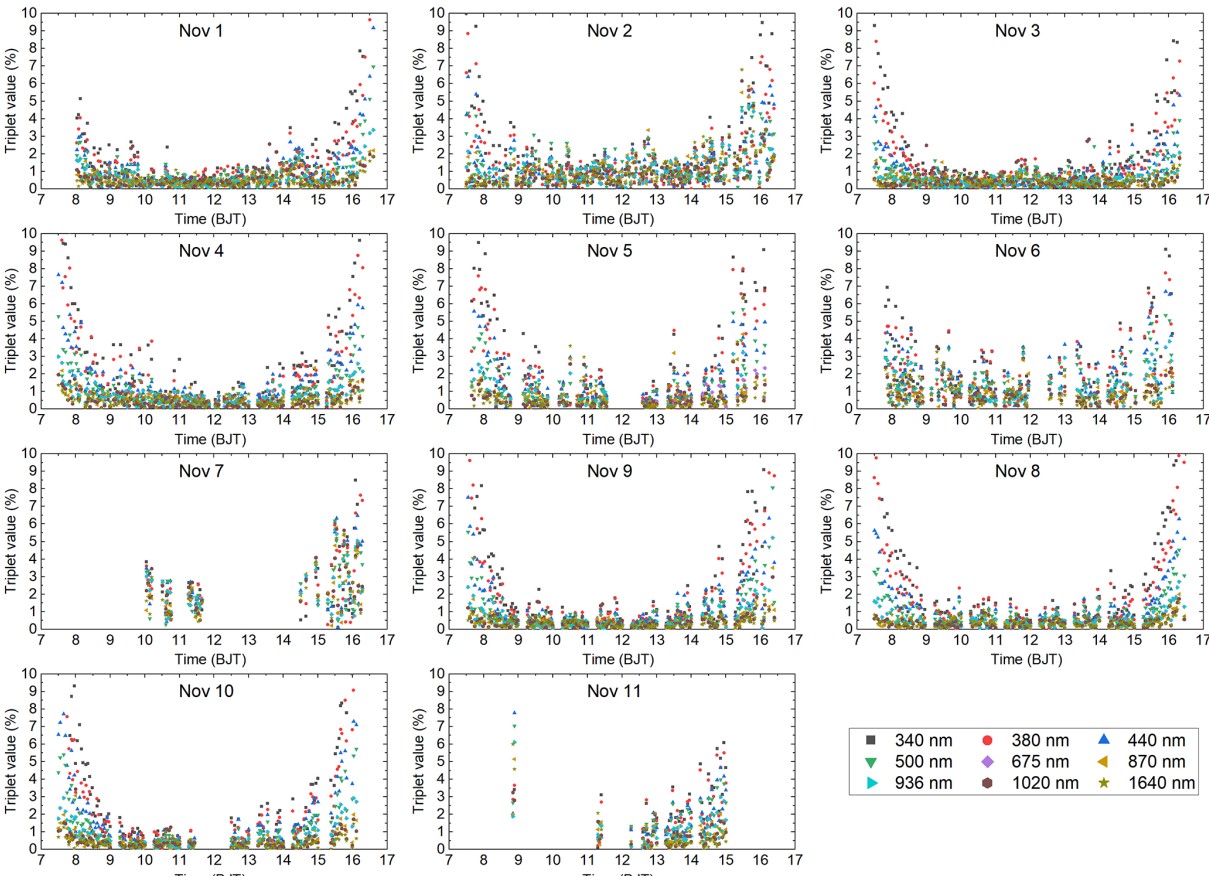

**Figure 4.** Diurnal variation of the triplet at each wavelength on 11 case days.

that on the other days, with almost all values exceeding 2.0 % at all times. The observation conditions on the other days were less affected by cloud, and it can be seen that the values reduced to a relatively low level; most values were < 2.0 % between 10:00 and 14:00 BJT (Beijing Local Time) in all cases. The triplets of the UV bands are as large as 10.0 %, whereas they are 2.0 %–6.0 % for the visible bands before 10:00 BJT and after 14:00 BJT. These results reveal that the digital counts of CW193 measurements fluctuate considerably during the morning and in the afternoon owing to the weak solar radiation and rapid and extensive changes in the solar altitudinal angle.

The daily average triplets were also calculated in this intercomparison (Fig. 5). We found that the daily average triplets for the UV bands showed the largest ranges of amplitude fluctuations: ∼ 1.5 %–3.0 % for 340 nm and 1.2 %–2.5 % for 380 nm. For the visible bands from 440 to 870 nm, it can be clearly seen that the variation in the daily average triplet decreases with increasing wavelength. With the exception of 7 November, which was greatly affected by cloud processes, the daily average triplets for the visible bands were all less than 2.0 %. Relatively weak fluctuation amplitudes were observed in the infrared bands from 1020 to 1640 nm

in all cases, with the daily average triplets being lower than 1.0 % except on 7 November, and they showed less variation with wavelength. The fluctuation for the WV channel at 936 nm was moderate compared with those for the other bands, and the daily average triplets were slightly higher than those in the infrared bands from 1020 to 1640 nm but much lower than those in the UV bands. In general, the WV channel showed a similar variation range to the 870 nm band: ∼ 0.5 %–2.5 %. As can be seen from Fig. 5, the lowest daily fluctuations were found on 3 November, with a variation range of ∼ 1.4 %–1.8 % for the UV bands and ∼ 0.4 %–0.8 % for the other bands. Using the meteorological and environmental records as a reference (no cloud contamination and daily $PM_{2.5} \sim 11 \, \mu m \, m^{-3}$; Table 4), these results indicate that the dispersion of diurnal triplets is quite small under clear and stable weather conditions, suggesting that CW193 is capable of reliable measurements.

## 3.2 AOD evaluation

The AOD performance of the CW193 was tested at the Beijing-CAMS site using CE318s as the reference, as this instrument has been widely verified under a wide range of

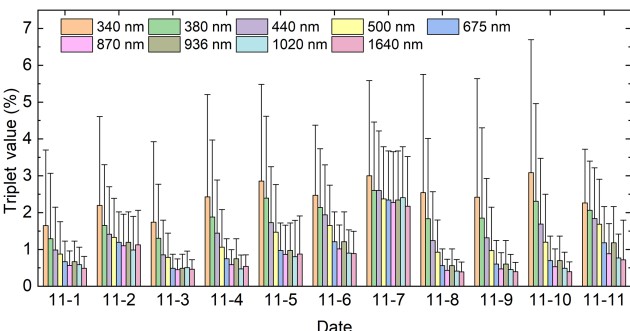

**Figure 5.** Daily triplet value at each wavelength on 11 case days from 1 Nov ("11-1") to 11 Nov ("11-11").

**Table 4.** Classification of case days based on daily average $PM_{2.5}$ and $PM_{10}$ concentrations and the variation range of $AOD_{440}$.

|  | Date | $PM_{2.5}$ | $PM_{10}$ | $AOD_{440}$ |
|---|---|---|---|---|
| Level I | 2 Nov | 6 | 42 | 0.08–0.15 |
|  | 3 Nov | 11 | 44 | 0.09–0.26 |
|  | 8 Nov | 12 | 45 | 0.11–0.21 |
|  | 1 Nov | 15 | 73 | 0.14–0.29 |
|  | 9 Nov | 23 | 57 | 0.14–0.31 |
|  | 7 Nov | 30 | 142 | 0.26–0.47 |
| Level II | 4 Nov | 37 | 77 | 0.35–0.58 |
|  | 10 Nov | 43 | 81 | 0.20–0.60 |
| Level III | 5 Nov | 82 | 125 | 0.49–0.91 |
|  | 6 Nov | 84 | 147 | 0.37–0.63 |
|  | 11 Nov | 104 | 148 | 1.32–1.47 |

conditions (Che et al., 2015, 2018; Holben et al., 2001; Xia et al., 2016).

First, we examined the wavelength dependence of the AOD from CW193, which is an important indicator of the observation precision. Furthermore, the daily average $PM_{2.5}$ and $PM_{10}$ concentrations were calculated for air quality classification, using the ambient air quality standards of China (GB3095-2012, http://www.mee.gov.cn/gkml/hbb/bwj/201203/t20120302_224147.htm, last access: 29 March 2022) as the reference, to achieve a comprehensive evaluation of AOD performance under different atmospheric pollutant loadings. In this study, Level I air quality is defined as a daily average $PM_{2.5}$ lower than $35\,\mu g\,m^{-3}$, which indicates that the air quality is quite clean and satisfactory for outdoor activities. Level II reflects acceptable air quality coincident with a low burden of certain air pollutants, and a daily average $PM_{2.5}$ concentration between 35 and $75\,\mu g\,m^{-3}$. Level III indicates mild atmospheric pollution with a daily mean $PM_{2.5}$ concentration of $75–115\,\mu g\,m^{-3}$. At Level III, the time spent on outdoor activities should be reduced for children, older people and patients.

The daily average $PM_{2.5}$ and $PM_{10}$ and the variation range of AOD at 440 nm ($AOD_{440}$) at different air quality levels are shown in Table 4. Figure 6 shows the diurnal variation of cloud-screened AOD (only from daytime observations) for each band from CW193 during this campaign. An obvious decreasing trend in AOD with increasing wavelength can be seen, which is in agreement with many previous studies (Che et al., 2019c; Holben et al., 1998; Liang et al., 2019). Consequently, under weak pollution conditions, the high AOD made the characteristics of the wavelength dependence more apparent. On the most polluted day (11 November, Level III, $PM_{2.5} \sim 104\,\mu g\,m^{-3}$, $AOD_{440} \sim 1.32–1.47$), the diurnal AOD was distributed in an orderly pattern and showed a similar variation trend at each wavelength, with each curve clearly visible and not intersecting with the others. This distribution was also found during the Level II situations on 4 and 10 November. Although $AOD_{440}$ ($\sim 0.20–0.60$) was smaller than that at Level III, the diurnal AOD curves for each wavelength were more continuous and could

be recognized more easily, which is partly attributable to the reduced impact of cloud contamination. In terms of AOD evaluation, the key point is that the performance under quite low aerosol loading is largely affected by the accuracy and stability of the instrument (Campanelli et al., 2007; Che et al., 2009; Kazadzis et al., 2018; Ningombam et al., 2019; Tao et al., 2014). From Fig. 4, it can be seen that (with the exception of 7 November, when severe cloud contamination occurred) the variation of the AOD curve can be easily identified owing to its wavelength dependence. Under the cleanest conditions (Level I, $PM_{2.5} \sim 6\,\mu g\,m^{-3}$, $AOD_{440} \sim 0.08–0.15$, 11 November), despite the cloud contamination in the afternoon, the AOD variation in each band consistently showed a gradually increasing trend, strictly following the rule of decreasing AOD with increasing wavelength. Therefore, in summary, CW193 showed good ability to reveal the wavelength dependence of AOD under both high and low aerosol loadings; hence, the excellent detection ability of CW193 makes it a reliable instrument for aerosol monitoring.

In the next step, the precision performance of CW193 was validated in detail using the AOD from AERONET as a reference. Figure 7 shows a comparison of the AODs from CW193 with the AODs from AERONET at coincident spectral wavelengths. In general, the AODs from CW193 agree well with the AERONET results, with correlation coefficients ($R$) of $\sim 1.000$ for 340–675 nm, $\sim 0.999$ for 870 nm, and $\sim 0.997$ for 1020 and 1640 nm, which indicates that the AODs from CW193 were similarly distributed on both sides of the $y = x$ line. From the $R$ values, we can see that the correlation tends to slightly decrease with increasing wavelength. This result can be explained to some degree by the temperature sensitivity of the instrument. As reported by Campanelli et al. (2007), the AOD in near-infrared bands is dependent on the ambient temperature, and the retrieval accuracy could be improved by correcting the data for the 870 and 1020 nm wavelengths for temperature effects. In addition, although CW193 is equipped with the same type of

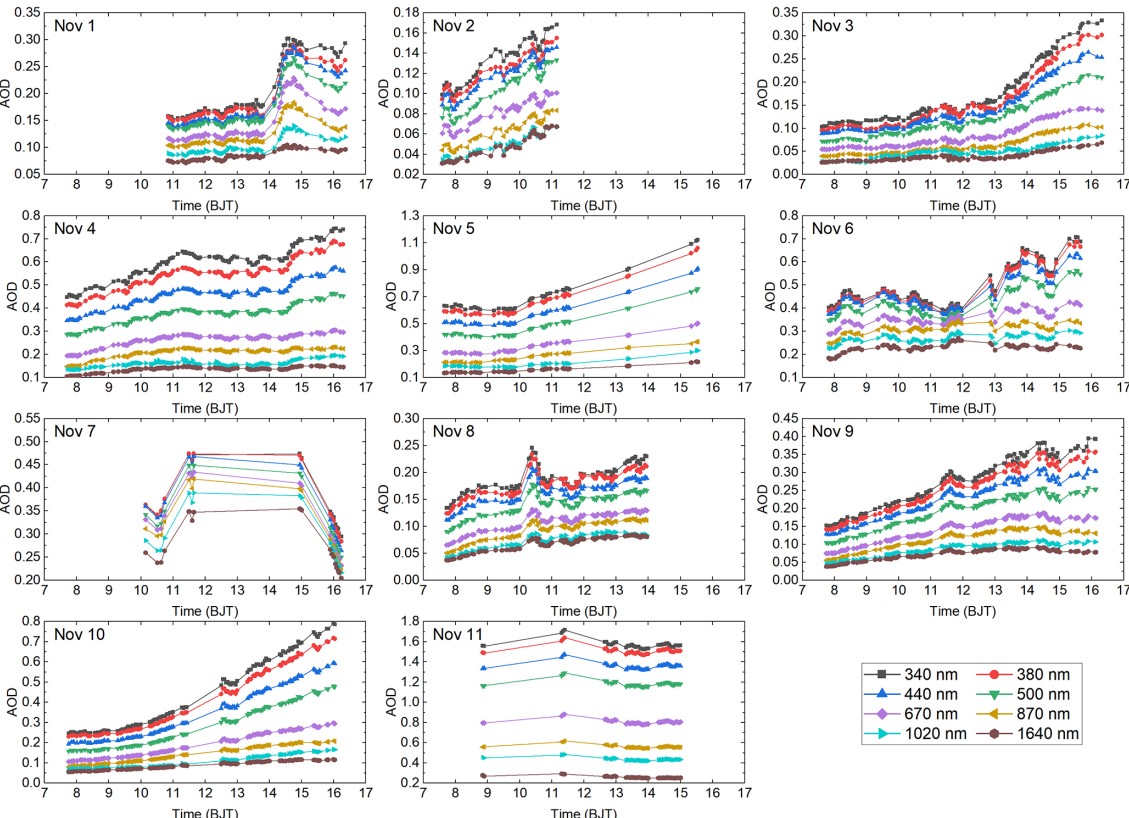

**Figure 6.** Diurnal variation of the AOD at each wavelength on 11 case days.

temperature sensor in its optical head, there are many other factors that influence the temperature readings, such as the mechanical structure and coating color, which could be the main reasons for the temperature uncertainty and the larger AOD deviations at the longer wavelengths of 870, 1020 and 1640 nm.

From this linear regression figure, it can be seen that the slopes for the 340 and 1020 nm bands are ~ 0.993 and 0.966, respectively, whereas those for the other bands are all larger than 1, varying from ~ 1.001 to 1.021. This indicates that the AOD from CW193 tends to be higher than that from AERONET. As done in many previous AOD validation studies, expected error (EE) analyses were conducted in this study. We set the envelopes as ± (0.05 + 10 %). It was found that the AODs from CW193 for each band were all able to achieve a satisfactory performance, with 100 % retrievals within the EE – much higher than the standard deviation of ~ 70 % (Che et al., 2019b; Levy et al., 2010). The root mean square errors (RMSEs) were all less than 0.05 for all bands, which revealed that the AODs from CW193 are all highly concentrated in the reference AOD range. These extremely small deviations also highlight the stability and accuracy of CW193. To further evaluate the AOD accuracy, the relative mean bias (RMB) for each linear regression equation was calculated. As mentioned above, the AOD

uncertainties for the near-infrared bands are obviously larger than those for the other bands in this campaign. Specifically, the AODs in the 1020 nm band were underestimated by ~ 7.8 % (RMB = 0.922), whereas they were overestimated by ~ 11.2 % (RMB = 1.112) in the 1640 nm band. The AODs from CW193 in the other bands were all slightly overestimated (~ 1.6 %–4.4 %), with the RMB varying within the relatively narrow range of ~ 1.016–1.044. This indicates that, from the perspective of stability and accuracy, the AODs derived from CW193 show better performance in the UV bands (340 and 380 nm) and visible bands (440 to 870 nm) than in the near-infrared band from 1020 to 1640 nm. Further studies and experiments aimed at algorithm and mechanical structure optimization to improve the retrieval accuracy need to be conducted in the future.

Figure 8 shows the CW193 AOD bias compared with equal-frequency bins of AOD from AERONET. All collocations of AODs were sorted in ascending order and then sampled with 20 bins. From the bias boxplots, it can be seen that the mean biases (red dots) have similar trends in the 340 to 870 nm bands: they are in a narrow range from about −0.02 to 0.03 and are characterized by an initial increase followed by a decrease and then a slight increase at high AOD. This indicates that the AODs in these bands from CW193 are overestimated at low AOD (for example, $AOD_{440} \sim 0.10$

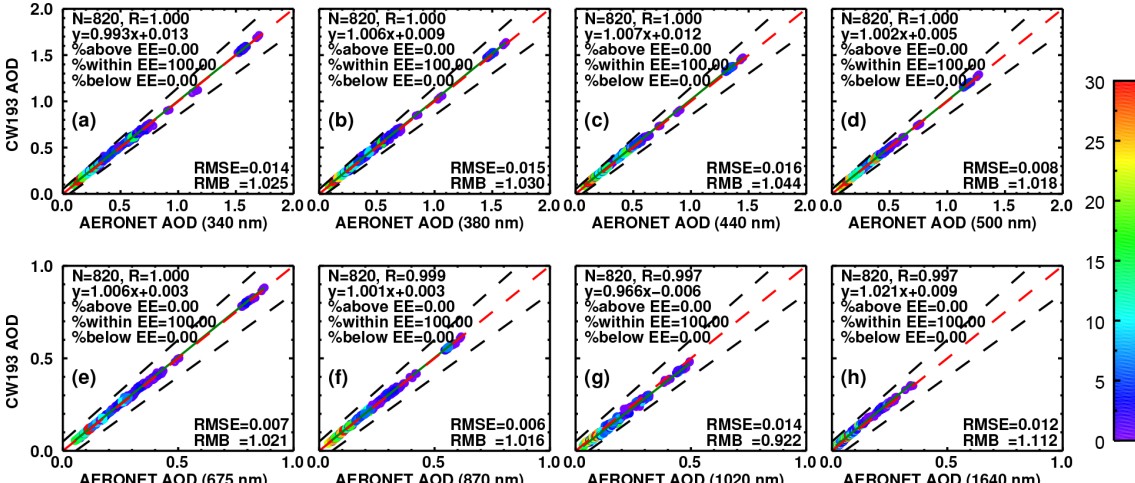

**Figure 7.** Validation of the CW193 AOD at each wavelength against the AERONET AOD. The one-to-one line, the linear regression line and the EE envelopes of $\pm(0.05 + 10\%)$ are plotted as dashed red, solid green and dashed black lines, respectively.

to 0.40). Then, under moderate AOD levels (for example, $AOD_{440} \sim 0.50$ to 0.90), these biases become smaller or almost equal to zero (or even a little bit negative) in the range from about $-0.01$ to 0.01, indicating that the calculations are more consistent with the reference values and are highly accurate. At high AOD levels (for example, $AOD_{440} \sim 1.30$ to 1.50), a slight increase in bias is observed in this campaign, with mean values varying from about 0 to 0.02. However, the bias performance is quite different for the 1020 and 1640 nm bands. For the 1020 nm band, the mean bias decreases from zero to $-0.02$, consistent with the AOD varying from $\sim 0.05$ to 0.20, and it remains relatively constant at about $-0.02$ as the AOD continually increases to $\sim 0.50$. For the biases at 1640 nm, the mean values in the bins show a roughly parabolic distribution, varying from $\sim 0.01$ to 0.02, consistent with the AOD varying from $\sim 0.02$ to 0.36. Similar to the results mentioned above, the different distribution of the bias boxes for the near-infrared bands suggests that an improvement in accuracy is needed. Although the linear regression and bias showed fluctuations to some degree, the AOD performance of CW193 was outstanding: high accuracy and stability were obtained based on the comprehensive analysis above, characterized by a bias concentrated within $\sim 0.02$ for the visible and near-infrared bands and within $\sim 0.03$ for the UV bands, which meets the accuracy requirements for AERONET (Holben et al., 1998).

## 3.3 Inversion evaluation

According to the algorithm, the aerosol inversions, including microphysical, optical and radiative properties, are retrieved from the aureole and sky radiance measurements. Similar to CE318, CW193 conducts the ALM routine at a specific time related to the air mass. It is performed in two wings in the 440, 675, 870 and 1020 nm bands sequentially: right

(azimuth angle displaced towards the right of the position of the sun) and left (azimuth angle displaced towards the left of the position of the sun). In this study, we chose the VSD, SSA and ADRF to represent the microphysical, optical and radiative properties of aerosols, as they are not only widely used parameters in current research but they are also the most important factors influencing the radiative budget of the earth–atmosphere system (Wang et al., 2013; Zhang et al., 2018). However, it should be noted that the uncertainties in these inversions are more difficult to ascertain, as the aureole and sky radiance measurements constitute only single observations (from one ALM routine), and the observation time of each sequence at a specific wavelength is largely dependent on the mechanical design and instrument version used (for example, the CE318-T TS11 mode CE8 has faster robotic movements than the CE318-N TS12 mode CE9). Furthermore, there is no absolute self-calibration procedure between the sphere calibrations; therefore, the uncertainty in the sky radiance at the time of calibration is assumed to be $< 5\%$ for these four channels (Holben et al., 1998). As reported by Tao et al. (2014), the sphere calibration results of CARSNET differed by $3.12\%$–$5.24\%$ in the 870 and 1020 nm bands and by less than 3 % in the other two bands compared with the original values from Cimel. In addition, to avoid disturbances from transient atmospheric processes, only the results with observation times that deviate by less than 10 min from those of AERONET were selected, and the related inversions of CARSNET were also retrieved and presented to show a more detailed comparison.

### 3.3.1 Volume size distribution

Figure 9 shows a comparison of the VSDs for four selected cases in this campaign. It can be seen that the results from CW193 accurately present the variation pattern

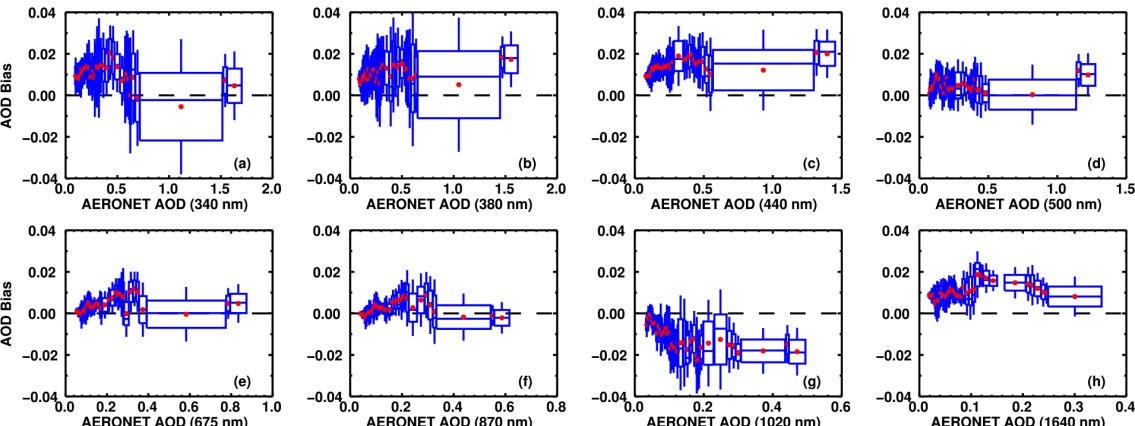

**Figure 8.** Boxplots of CW193 AOD bias and AERONET AOD using the 25th and 75th percentiles with 20 bins. The dashed black line indicates zero bias. The red dots, middle line, and upper and lower hinges represent the mean and median of the AOD bias and the 25th and 75th percentiles, respectively.

of aerosols: typical bimodal distributions were seen on 6 and 10 November and nearly unimodal distributions for the two cases on 7 November. For fine-mode particles (radius $< 1.00\,\mu\text{m}$), variations were clearly observed on 6 and 10 November. For the reference PM concentrations, the ratio $PM_{2.5}/PM_{10}$ was $\sim 53.1\,\%\text{–}57.1\,\%$, suggesting a certain amount of small particles, which agrees with the distribution pattern from CW193 and AERONET. The maximum volume of fine-mode particles varied in the range of $\sim 0.03\text{–}0.05$ and $\sim 0.07\text{–}0.08\,\mu\text{m}^3\,\mu\text{m}^{-2}$ on 6 and 10 November, respectively. Specifically, the largest deviations of the maximum for fine-mode particles occurred on 6 November: $\sim 77.6\,\%$ and $\sim 57.1\,\%$ for CW193 and CARSNET compared with AERONET, respectively. Despite the large volume deviations for fine-mode particles, the variation trends were consistent with those of AERONET, characterized by a maximum peak at a radius of $0.15\,\mu\text{m}$. Hence, these patterns can be attributed to the different observation times to some degree. The time deviation compared with AERONET varied from $\sim 3$ to $4\,\text{min}$ in this case, while the fine-mode volume showed a gradually decreasing trend from CW193 to CARSNET to AERONET, which agreed with the time series. In contrast, small deviations of the maximum for fine-mode particles occurred on 10 November: $\sim 8.9\,\%$ and $\sim 6.8\,\%$ for CW193 and CARSNET compared with AERONET, respectively. The peaks for CW193 and AERONET occurred at a radius of $0.11\,\mu\text{m}$ and the peak for CARSNET was observed at $0.15\,\mu\text{m}$, indicating that both CW193 and CARSNET show good consistency with AERONET.

For coarse-mode particles (radius $> 1.00\,\mu\text{m}$), variations were clearly detected for the four cases, especially on 7 November, when the ratio $PM_{2.5}/PM_{10}$ was $\sim 21.1\,\%$, suggesting that large aerosols were dominant. In these four cases, the peak volumes of coarse-mode particles varied in the ranges $\sim 0.09\text{–}0.13$, $\sim 0.11\text{–}0.14$, $\sim 0.18\text{–}0.25$ and $\sim 0.05\text{–}0.07\,\mu\text{m}^3\,\mu\text{m}^{-2}$, respectively. It can be seen that the

highest deviations of the peak volume from CW193 were observed on 6 and 7 November, with values of $\sim 29.2\,\%$, $\sim 19.1\,\%$ (for the case corresponding to a time of around 08:00) and 22.2 % (for the case corresponding to a time of around 12:00) compared with AERONET, respectively. However, the performance of CARSNET was better than that of CW193 in these three cases, with deviations of $\sim 5.7\,\%$, $\sim 20.4\,\%$ and $\sim 6.7\,\%$, respectively. As mentioned above, except for the calibration and algorithm uncertainties, these large deviations could be explained by the influences of instantaneous atmospheric disturbances on the retrievals, although the time differences between the CW193 and AERONET measurements were within $\sim 4\text{–}8\,\text{min}$ ($\sim 3\text{–}4\,\text{min}$ for CARSNET). A narrow variation range was found for the 10 November case, which was characterized by a relatively small time difference among these three retrievals ($\sim 2\text{–}4\,\text{min}$). Consequently, the deviation of the peak volume for CW193 compared with AERONET was $\sim 13.1\,\%$, while a larger difference of $\sim 16.8\,\%$ was found for CARSNET. In summary, the difference in the VSD showed an obvious time-correlation regularity – the smaller the time deviation, the better the retrieval consistency with AERONET.

### 3.3.2 Single scattering albedo

The SSA represents the scattering proportion affected by aerosol particles with respect to the total extinction, and is one of the key variables for assessing the effects of aerosols on the climate (Che et al., 2019c; Zhao et al., 2018). The variation of the SSA at four spectral wavelengths for the four cases (6 and 10 November and two on 7 November) is shown in Fig. 10. First, we examined the wavelength dependence of SSA, revealing the different scattering capacities for aerosols at specific bands, which is largely influenced by the aerosol chemical composition and can be regarded as an indicator of the dominant aerosol type (Eck et

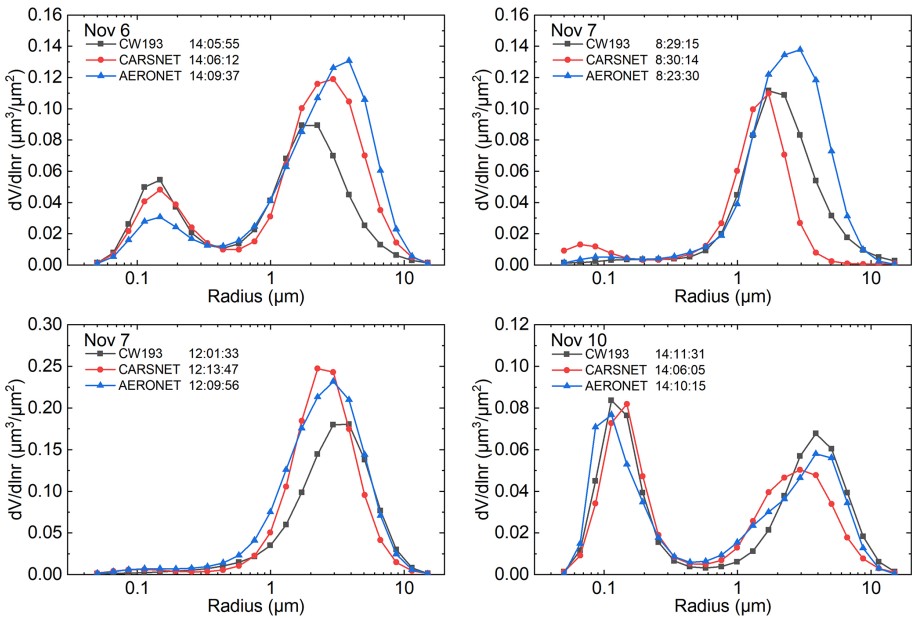

**Figure 9.** Comparison of retrieved VSDs from CW193, CARSNET and AERONET for four selected cases.

al., 1999; Zheng et al., 2021). It can be seen from Fig. 10 that the SSA showed different variation trends on the 3 days. Specifically, for the 6 November case, the SSA increased from 440 to 675 nm and showed a roughly decreasing trend from 675 to 1020 nm, indicating a relatively strong aerosol absorbance at shorter wavelengths in the visible bands. The SSA showed an increasing trend with wavelength for the two cases on 7 November, whereas a decreasing trend was observed on 10 November. This indicates that the absorptive ability of the aerosol was attenuated with increasing wavelength on 7 November, whereas enhanced aerosol absorbance with wavelength occurred on 10 November. From the discussion above, we can see that the wavelength dependences of the SSAs from CW193 and CARSNET were both highly consistent with that from AERONET, indicating that this retrieval showed good performance for aerosol optical properties.

To elaborate the SSA assessment, we present a comprehensive comparison of the accuracy in detail here. On 6 November, the SSA peaked in the 675 nm band, with values of $\sim 0.848$, 0.857 and 0.853 for CW193, CARSNET and AERONET, respectively. The deviations of these maxima for CW193 and CARSNET compared with AERONET were $\sim 0.1\,\%$ and 0.3 %, respectively. In this case, the SSA of CW193 varied within a narrow range of $\sim 0.834$–0.848, whereas that of AERONET was $\sim 0.836$–0.853. The highest deviation for a specific wavelength of CW193 was found in the 1020 nm band, with a value of $\sim 1.7\,\%$, and the lowest was found in the 440 and 870 nm bands, with a value of $\sim 0.1\,\%$. As mentioned above, the SSA showed an increasing trend with wavelength for the two cases on 7 November. The smallest SSA values were all observed in the 440 nm

bands, with values varying in the range of $\sim 0.858$–0.861 and $\sim 0.840$–0.859, respectively. For the case at around 08:00, the maximum of CW193 was found in the 870 nm band, with a value of $\sim 0.899$, whereas that of AERONET was found in the 1020 nm band, with a value of $\sim 0.911$, which suggests a maximum deviation of $\sim 1.3\,\%$. The largest deviation for a specific wavelength of CW193 compared with AERONET occurred in the 1020 nm band, and was $\sim 2.1\,\%$, followed by 1.9 % at 675 nm, 1.0 % at 870 nm and 0.6 % at 440 nm. For the case at around 12:00, although the SSA was relatively low in the 440 nm band ($\sim 0.840$–0.859), it remained almost constant from 675 to 1020 nm for CW193 and AERONET; it was characterized by a small fluctuation amplitude of $\sim 0.930$–0.935 for the former and $\sim 0.926$–0.931 for the latter. The highest deviation for a specific wavelength, $\sim 1.8\,\%$, was measured in the 440 nm band, followed by 0.8 % at 1020 nm, 0.6 % at 675 nm and 0.1 % at 870 nm. The SSA showed more obvious fluctuations for the 10 November case. Specifically, the peak SSAs from CW193 and AERONET were both observed in the 440 nm band, with values of $\sim 0.844$ and 0.832, respectively. Likewise, the lowest values of $\sim 0.733$ and 0.708 for these two were measured in the 1020 nm band. However, the variation of the deviation at a specific wavelength did not show a regular pattern compared with the SSA. The largest deviation of $\sim 3.5\,\%$ was found in the 1020 nm band, followed by $\sim 2.6\,\%$ at 870 nm, $\sim 1.4\,\%$ at 440 nm and $\sim 0.7\,\%$ at 765 nm. In conclusion, the SSA deviation between CW193 and AERONET varied in the range of $\sim 0.1\,\%$–1.8 %, $\sim 0.6\,\%$–1.9 %, $\sim 0.1\,\%$–2.6 % and $\sim 0.8\,\%$–3.5 % for the 440, 675, 870 and 1020 nm bands, respectively, indicating a high consistency with AERONET.

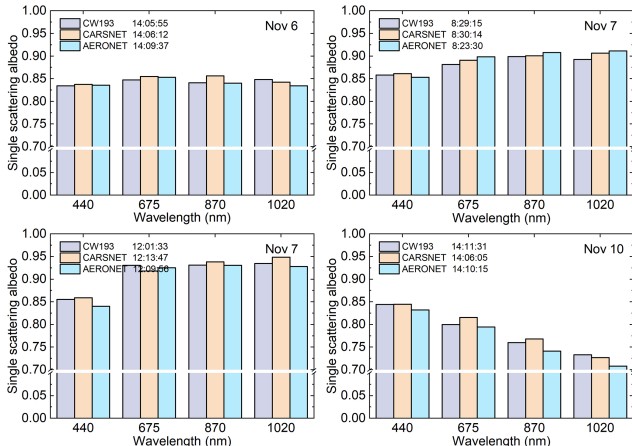

**Figure 10.** Comparison of the retrieved SSAs from CW193, CARSNET and AERONET for four selected cases.

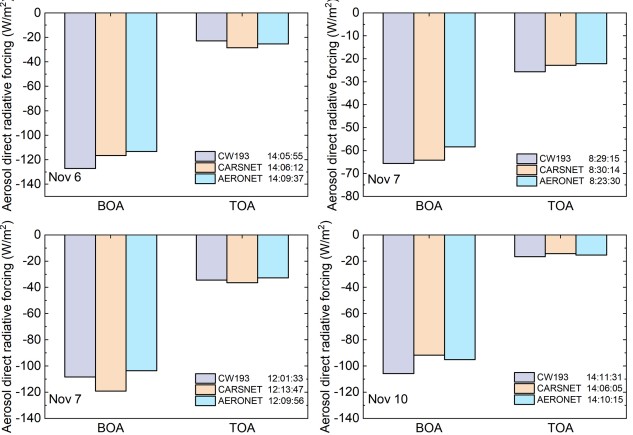

**Figure 11.** Comparison of the retrieved ADRF values from CW193, CARSNET and AERONET for four selected cases.

### 3.3.3 Aerosol direct radiative forcing

The ADRF is a key factor influencing the radiation budget of the earth–atmosphere system; any small perturbation to this global energy balance can cause a profound change in the climate (García et al., 2012). In this context, much progress had been made in this field to provide insight into the climate effects of aerosols. A previous study estimated the total anthropogenic radiative effect at the global scale to be +1.6 (−1.0 to +0.8) W m$^{-2}$, of which −0.5 (±0.4) W m$^{-2}$ are associated with the direct radiative forcing of aerosols (García et al., 2008). However, it can be seen that there remains huge uncertainty in the evaluation of the ADRF. For this reason, we selected this important product of CW193 to examine the accuracy of the radiative retrieval.

In Fig. 11, we show a comparison of the ADRF values from CW193, CARSNET and AERONET for the four cases (6 and 10 November and two on 7 November). As reported by Zheng et al. (2019), the ADRF at the earth's surface

(BOA) varies from −86 ± 31 to −132 ± 50 W m$^{-2}$, whereas the ADRF at the top of the atmosphere (TOA) varies from −35 ± 18 to −55 ± 26 W m$^{-2}$, based on a 5-year observation campaign in urban Beijing. Therefore, it can be seen that the BOA and TOA retrievals of CW193 and CARSNET all show a reasonable range of values in this campaign. Specifically, the BOAs of CW193 were −127.1, −65.6, −108.4 and −105.6 W m$^{-2}$ for the four cases in chronological order, respectively. Correspondingly, the BOAs from AERONET were −113.2, −58.4, −103.5 and −95.0 W m$^{-2}$. Thus, the deviation of the BOA in these cases was ∼ 12.2 %, 12.3 %, 4.8 % and 11.2 %, respectively, suggesting an overestimation of BOA compared with AERONET. For the TOAs, the CW193 retrievals for these cases were −22.8, −25.6, −34.3 and −16.5 W m$^{-2}$, whereas the reference values from AERONET were −25.3, −22.1, −32.6 and −15.3 W m$^{-2}$, respectively. That is, the TOA deviation found in these cases was ∼ 9.8 %, 15.9 %, 5.4 % and 7.4 %, respectively. In summary, the deviation of the retrieval BOA was ∼ 5 %–12 %, whereas it was ∼ 5 %–16 % for the TOA. As shown above, the relatively large uncertainties can be partly explained by the inherent algorithm error, as well as the difference in observation time.

### 3.4 Water vapor evaluation

Water vapor (WV) is a key atmospheric component in studies of climate change because it not only has an important role in aerosol aging but it can also influence the energy budget of the earth–atmosphere system by absorbing and scattering solar energy. Therefore, in this study, the precision performance of WV from CW193 was validated in detail using AERONET as a reference.

Figure 12 shows a comparison of the WV from CW193 with the results from AERONET. In Fig. 12a, it can be seen that the WV from CW193 agrees well with the AERONET WV, with a correlation coefficient (R) of ∼ 0.997. From this linear regression, the slope was ∼ 0.941, suggesting that the WV from CW193 tends to be lower than that from AERONET. In terms of RMB values, it is found that the WV from CW193 is underestimated by ∼ 2.1 % (RMB = 0.979). EE analysis showed that the retrieved columnar WV (100 %) was within the EE. In addition, the small RMSE (∼ 0.020) also indicated that the CW193 WV was highly concentrated in the reference AERONET range.

Figure 12b shows the CW193 WV bias compared with equal-frequency bins of WV from AERONET. From this boxplot, it can be seen that the bias varies in the range of −0.04 to 0.04, whereas its mean values (red dots) are concentrated in a narrower range from −0.02 to 0.02. As reported by Holben et al. (1998), the uncertainty of the WV retrieval has been found to be less than 12 %, based on an intercomparison with radiosonde results. In this study, the overall WV bias of CW193 was roughly lower than 4 %, demonstrating the accurate measurement capability of CW193 for columnar

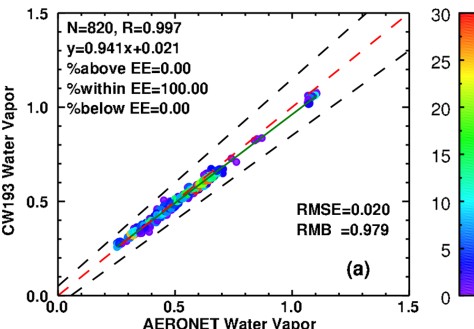 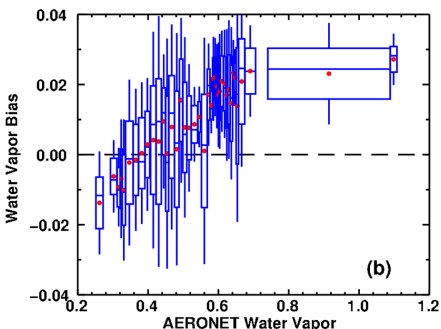

**Figure 12.** As in Figs. 7 and 8, but for water vapor (unit: cm).

WV. However, it should be noted that the bias, especially its mean value, shows an increasing trend (from about $-0.01$ to 0.03) with increasing WV (from $\sim 0.24$ to 0.80 cm). Gui et al. (2017) revealed that the monthly WV for November was $\sim 0.74$ cm in urban Beijing, whereas that for the summer exceeded 2.00 cm. In this campaign, the CW193 WV varied from $\sim 0.26$ to 1.08, indicating that, in the future, it will be necessary to test whether this increasing trend in the bias actually exists, especially on humid summer days.

## 4 Conclusions

In this study, we have presented a multiwavelength photometer named CW193 for monitoring aerosol microphysical, optical and radiative properties. The CW193 is highly integrated and is composed of three main parts: the optical head, robotic drive platform and stents system. It has a user-friendly interface, and all commands can be sent to the instrument via serial communication or the 4G network, which makes data acquisition and operation monitoring easier. A performance evaluation of CW193 based on an intercomparison with the reference AERONET results was presented and discussed in detail. The main conclusions of this study are as follows:

1. The comparison of raw digital counts from CW193 and CE318s (two AERONET master instruments, photometers #1043 and #1046) showed high coefficients of determination ($R^2$) for all wavelengths: $> 0.97$ and $> 0.99$, respectively. Apart from when there was cloud contamination, the diurnal triplets for these nine bands were mostly lower than 2.0 % during 10:00 to 14:00 BJT. Daily average triplets for the UV bands (340 and 380 nm) varied from about 1.2 % to 3.0 %, whereas they were $< 2.0$ % for the visible and infrared bands (440–1640 nm).

2. Using reference PM concentrations, the wavelength dependence of the AOD was examined. The AOD curves did not intersect and were easily identified (AOD$_{440} \sim$ 0.08 to 1.47) for Level I to Level III air quality (PM$_{2.5} \sim$

6 to 104 µg m$^{-3}$), and they showed a decreasing trend with increasing wavelength. From the regression analysis, good AOD agreement ($R > 0.99$) and RMSE values ranging from 0.006 (870 nm) to 0.016 (440 nm) were observed. CW193 showed satisfactory AOD performance, with 100 % of the retrievals lying within the EE ($0.05 + 10$ %), and the RMB varying from 0.922 to 1.112. The AOD bias analysis showed an overall deviation of within $\pm 0.04$, and the deviation in the mean value was within 0.02.

3. The variation of inversions was dependent on the time of the measurement in this study. From the perspective of VSD retrievals, the deviation of the maximum for fine-mode particles varied from $\sim 8.9$ % to 77.6 %, whereas it varied from $\sim 13.1$ % to 29.1 % for coarse-mode particles. The wavelength dependence of the SSA from CW193 showed a similar trend to the AERONET SSA, and the variation range of the deviation was $\sim 0.1$ %– 1.8 %, $\sim 0.6$ %–1.9 %, $\sim 0.1$ %–2.6 % and $\sim 0.8$ %– 3.5 % for the 440, 675, 870 and 1020 nm bands, respectively. For the ADRF, BOA and TOA, deviations of $\sim 4.8$ %–12.3 % and $\sim 5.4$ %–15.9 %, respectively, were observed in this study.

4. Good WV agreement was found, characterized by a high $R$ ($\sim 0.997$), a small RMSE ($\sim 0.020$) and a satisfactory EE distribution (100 % within the EE). The RMB showed that the WV was underestimated by $\sim$ 2.1 % (RMB $= 0.979$). The bias mostly varied within $\pm 0.04$ cm, whereas its mean values were concentrated within $\pm 0.02$ cm.

The results of this preliminary evaluation indicate that CW193 is appropriate for monitoring aerosol microphysical, optical and radiative properties, with the overall AOD (including WV) bias within $\pm 0.02$ for the 500 to 870 nm bands and within $\pm 0.04$ for the other bands. Considering the uncertainty inherent in the algorithm ($\pm 0.02$) and the AOD uncertainty of AERONET ($\pm 0.02$), the direct sun measurements seem reasonable and reliable for the AOD and WV calculations (uncertainty within $\pm 0.04$). However, its perfor-

mance under extreme heavy aerosol loading, especially during severe haze and/or dust episodes when the AOD exceeds 2.00, still needs to be assessed. Although the results for the SSA and ADRF showed good agreement with those from AERONET, the VSD deviations were larger than for those two parameters. In fact, owing to the joint influence of the sphere calibration uncertainty and the measurement time difference, evaluating these inversions for the short period of the observation campaign was difficult. Consequently, the instruments need to be further tested under different environmental conditions, including via long-term observations in mountainous, coastal and desert regions with CE318, POM-02 and PFR used as references. As a result, the CW193 retrievals in this study showed high precision for SSA and ADRF and comparable results for VSD, indicating good comparability and consistency with AERONET.

Above all, the highly integrated design and smart control performance of CW193 make it suitable for monitoring the microphysical, optical and radiative properties of aerosol. Due to its smart control performance and optional observation schedule, such as the ALM mode, CW193 can meet the different requirements for aerosol microphysical, optical and radiative properties. When the VSD and SSA are in great demand to aid the modification of numerical models and the verification of satellite inversion products, these inversions can be obtained about 2–3 times in an hour, or once per hour in the default observation schedule. As a result, this instrument could be regarded as a contributor to regional and climate model data assimilation, satellite modification, and improving knowledge of the temporal and spatial variations of aerosols.

*Data availability.* Datasets used in the present study are available from the corresponding author on reasonable request.

*Supplement.* The supplement related to this article is available online at: https://doi.org/10.5194/amt-15-1-2022-supplement.

*Author contributions.* HC and XZ designed the research. HC, YW, XH and XZ built the device. XX and JZ performed the calculation. JZ, HZ and KG analyzed the data. YZ and LL wrote the paper. All authors discussed the results and commented on the paper.

*Competing interests.* The contact author has declared that neither they nor their co-author has any competing interests.

*Acknowledgements.* We are grateful to the AEROENT team for their assistance with the reference results used in this paper. We would like to acknowledge the master photometer calibration performed by Group of Atmospheric Optics (http://goa.uva.es/, last access: TS13) for Beijing-CAMS site of AEROENT. The authors would like to thank the five anonymous reviewers for their constructive suggestions and comments.

*Financial support.* This research has been supported by the National Science Fund for Distinguished Young Scholars (grant no. 41825011), the National Natural Science Foundation of China project (grant no. 42030608, 42105138, 41975161, 41905117, 42175153), the Basic Research Fund of CAMS (grant no. 451336) and the Science and Technology Development Fund of CAMS (grant no. 2019KJ001).

*Review statement.* This paper was edited by Cheng Liu and reviewed by five anonymous referees.

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

**Remarks from the language copy-editor**

CE1     What does "AOP" stand for? Please explain the abbreviation "AOP" somewhere in your paper, e.g. by inserting "(AOP)" in the abstract, if this should refer to "aerosol microphysical, optical and radiative properties". As this is no commonly known abbreviation, it needs an explanation.

CE2     Please confirm or correct this changed footnote. The question was not about the exact name of the photometer, but if the table shows parameters/data of the standard version/model of the photometer (as opposed to other versions of the device) or if the parameters/data in the table refer to a standard mode (as opposed to other specific modes). Please do not confuse "mode" with "model", see other comments.

CE3     Should this be "model", because this refers to the version/generation of the device? See next CE comment for explanation.

CE4     If CE318-N and CE318-T are technical devices of a different generation, than these should be "models" not "modes". Please confirm the changes in this sentence. Different "modes" would be something you can choose within one device, e.g. a digital camera has several modes you can choose, like 'macro' or 'black and white'. Please check comments where change from "mode" to "model" is suggested and please check yourself where else in the paper "mode" should be "model".

CE5     Please check and confirm/correct changed footnote a. Does "version" refer to a specific generation of a device (like in the footnote now), or should this refer to a mode (here 'standard mode') within a device? Please do not confuse "mode" and "model".

CE6     This has been changed to "0.9985" in accordance with the previous value. If you intend to change this to "0.9995" this requires the editor's approval (please write an explanation here, why this change has to be made).

CE7     Has been changed to "their" (the bands') sensitivity. Please confirm.

CE8     Should this be "model", because this refers to the version/generation of the device?

CE9     Should this be "model", because this refers to the version/generation of the device?

**Remarks from the typesetter**

TS1     Please write an explanation here, why this change has to be made. We have to send this request to the editor.

TS2     Please write an explanation here, why this change has to be made. We have to send this request to the editor.

TS3     Editor approval needed.

TS4     Editor approval needed.

TS5     Editor approval needed for changing "CE318-T" to "CE318-TS".

TS6     Editor approval needed.

TS7     Editor approval needed.

TS8     Editor approval needed.

TS9     Editor approval needed.

TS10     Editor approval needed.

TS11     Editor approval needed.

TS12     Editor approval needed.

TS13     Please provide your last access date.