# Peer review of "Evaluation of aerosol microphysical, optical and radiative properties measured from a multiwavelength photometer"

_Atmospheric Measurement Techniques, 2021_

## Author Comment (AC1)

**Referee #1**

The manuscript "A new multispectral photometer for monitoring aerosol microphysical, optical, and radiative properties" mainly describes a new multispectral photometer (CW193) proposed in this study. In this study, the design of multispectral photometer combines the merit of low maintenance requirements and being appropriate for the deployment in remote and unpopulated regions. In general, the paper is well written and presented in a logical way. It is a timely and important piece of work, and of general interest for Atmospheric Measurement Techniques related communities. I therefore recommend publication of this paper in Atmospheric Measurement Techniques after minor revisions. My comments are listed as follows:

**Response:** Thank you for giving us the opportunity to improve the quality of this manuscript. We have substantially revised this manuscript by following your insightful comments and constructive suggestions. Please find out our point-by-point responses below. We have studied comments carefully and have made correction which we hope meet with approval. Revised portion are marked in **red** in the revised paper.

Specific Comments:

1. Line 109-110, In the sentence of "the main pollution sources are derived from urban activities", the meaning of "source" has been already included in the word "derive".

**Response:** Thanks for pointing out. We have deleted this "source" in our manuscript.

*Lines 109-110 in the revised paper:*
*"...where the main pollution are derived from urban activities."*

2. Line 111, a description for is needed.

**Response:** Thanks for your suggestion. According to the comments, we guess a description for "CAMS" is needed in there. We added the full name of CAMS in our revised paper.

*Line 101 in the revised paper:*
*"...according to long-term ground-based aerosol measurements at CAMS (Chinese Academy of Meteorological Sciences) ..."*

3. In figure 1, I suggest the authors add the map of China as well as the CAMS location. Otherwise, the readers cannot catch the location information of CAMS.

**Response:** Thanks for your constructive comment. We have re-plotted this figure as to show the location of CAMS more clearly.

*Line 120 in the revised paper:*

[Figure]

4. Line 141, the role of word "respectively" is indistinct in the sentence.

**Response:** Thanks for your kind suggestion. In this sentence, we try to explain that the AOD is calculated from the Sun radiation measurements and the other microphysical, optical, and radiative properties of aerosols is retrieved from sky radiation measurements. We have rewritten this sentence as follow to make this explanation more accurate.

*Lines 141-142 in revised paper:*
*"The CW193 is an automatic photometer and designed to obtain AOD and other retrievals (such as microphysical, optical, and radiative properties of aerosols) from Sun radiation and sky radiation monitoring."*

5. Line 204, Is the meaning of same as?

**Response:** Thanks for your suggestion. We check through this part in our paper, and guess that the additional explanation is needed for the method of Sun calibration of CW193. As usual, there are two main calibration method for the sun radiance—Langley plot method and coefficient transfer method. For the AEROENT, the master instruments are calibrated at Mauna Loa Observatory (3397 m a.s.l) and Izaña Observatory (2373 m a.s.l) via Langley plot method, and then the calibration coefficient is transferred to field instruments by inter-comparison. As for the CARSNET, its master instruments are calibrated by the Group of Atmospheric Optics (GOA, in Valladolid, Spain) at Izaña for every six months. In this campaign, the CW193 (could be regarded as field instrument) was calibrated via coefficient transfer method (inter-comparison) with the reference of AERONET master instruments according to the Eq. 1 as below,

$$C(\lambda) = C(\lambda)_0 \times \left(\frac{V(\lambda)}{V(\lambda)_0}\right) \dots \dots Eq.\,1$$

where the $C(\lambda)$ and $C(\lambda)_0$ is the calibration coefficient for field instrument and master instrument at $\lambda$ wavelength, respectively. $V(\lambda)$ and $V(\lambda)_0$ is the digital count for field instrument and master instrument at $\lambda$ wavelength, respectively. We have rewritten this sentence

in paper and added one corresponding reference of coefficient transfer method to make it more accurate.

*Lines 204-205 in revised paper:*
*"...using the method of coefficient transfer (inter-comparison) with the reference master instruments of AERONET (Che et al., 2009, 2019c; Zheng et al., 2021)."*

*Che, H., Zhang, X., Chen, H., Damiri, B., Goloub, P., Li, Z., Zhang, X., Wei, Y., Zhou, H., Dong, F., Li, D. and Zhou, T.: Instrument calibration and aerosol optical depth validation of the China Aerosol Remote Sensing Network, J. Geophys. Res. Atmos., 114(D3), doi:10.1029/2008JD011030, 2009.*

6. Line 301 and Table 4, What is the standard of Level I-III? The corresponding information is needed.

**Response:** Thanks for your suggestions. In fact, we have already introduced the standard of Level I-III, and its classification is based on the ambient PM2.5 concentrations according to the ambient air quality standards of China (GB3095-2012, http://www.mee.gov.cn/gkml/hbb/bwj/201203/t20120302_224147.htm) in Lines 291-297. Briefly, Level I means the daily average $PM_{2.5} < 35$ μg m$^{-3}$, and the Level II reflects the $PM_{2.5}$ concentration between 35 μg m$^{-3}$ and 75 μg m$^{-3}$, while the Level III indicates the daily average $PM_{2.5}$ between 75 μg m$^{-3}$ and 115 μg m$^{-3}$.

7. In the bottom description of Figure 7, the sentence "One–one line, linear regression line, and the EE envelopes of ±(0.05 + 10%) are plotted as red dashed, green solid, and black dashed lines" should be changed to "One–one line, linear regression line, and the EE envelopes of ±(0.05 + 10%) are plotted as red dashed, green solid, and black dashed lines, respectively".

**Response:** Thanks for your constructive suggestions. It has been corrected.

*Lines 204-205 in revised paper:*
*"Figure 7. Validation of CW193 AOD at each wavelength against AERONET AOD. One–one line, linear regression line, and the EE envelopes of ± (0.05 + 10%) are plotted as red dashed, green solid, and black dashed lines, respectively."*

8. In the calculations of ADRF for CW193 and instruments of CARSNET, and AERONET, does the authors use the same radiation transfer model? If the model is different, the difference of ADRF may not induced by the instrument alone.

**Response:** Thank you so much for your constructive comments. In this study, the ADRF was calculated by the radiative transfer module, which is similar to the inversion of AERONET (García et al., 2008, 2012). In our revised paper, we have re-organized the section 2.2.3 to present the data processing method in this campaign, including AOD, WV, VSD, SSA, ADRF and their uncertainties. In fact, these retrievals uncertainties (VSD, SSA and ADRF) are greatly affected by the calibration processing, because there is no absolute self-calibration procedure between the sphere calibration,

indicating the differences of retrievals were joint determined by many factors, such as uncertainties of inherent algorithm assumption, input direct and sky radiance, surface albedo. As the results, in order to reduce the uncertainty from input radiance, we only used the results, which the observation interval is within 10 minutes to conduct the comparison. Through there are still some differences, we suggested that these results were comparable with the AERONET. In next step, we will further test its stability and accuracy based on long-term observation campaign, with the reference of AERONET results.

*Lines 214-229 in revised paper:*

*"We calculated the cloud-screened AOD and columnar water vapor of CW193 via the similar algorithm as AERONET. As the algorithm has been used multiple times in many observation campaigns, numerical modeling, and satellite verification for CARSNET, it is suitable and reliable to evaluate the AOD performance of CW193 using this method (Wang et al., 2010; Xia et al., 2021; Yu et al., 2015; Zhao et al., 2021c; Zheng et al., 2021). The algorithm verification is provided in the Supplementary Information to guarantee the accuracy in this campaign (Figures S1 and S2). As for the inversions of VSD and SSA in this campaign, they were retrieved from the observational data from the diffuse-sky measurements of the CW193 at 440, 670, 870, and 1020 nm using the algorithms of Dubovik et al. (2002, 2006). The ADRF was calculated by the radiative transfer module, which is similar to the inversion of AERONET (García et al., 2008, 2012). Because the introduction, validation and application of these inversions and their algorithms have been presented in many previous studies based on CARSNENT observation, we did not repeat these again in this paper (Che et al., 2018, 2019c; Zhao et al., 2018; Zheng et al., 2021). In general, the AODs' uncertainty was 0.01 to 0.02 (Eck et al., 1999). The VSD accuracy was 15 % to 25 % between 0.1 µm ≤ r ≤7.0 µm while 25 % to 100 % for other radius (Dubovik et al., 2002). The SSA accuracy was 0.03 when its was calculated under the condition of AOD440 nm >0.50 with a solar zenith angle >50 ° (Dubovik et al., 2002). The bias for measured radiation at the surface was about 9±12 W m$^{-2}$, affected by the dominant aerosol type (García et al., 2008)."*

---

## Author Comment (AC2)

**Referee #2**

This paper presents a newly-designed sun photometer for aerosol retrieval. Compared to the widely used CE318 model, the new instrument has the advantage of better portability with similar accuracy. Inter-comparisons are carried out to evaluate the performance, which shows that the CW193 sunphotometer has comparable retrieval accuracy. The new instrument has the potential to be deployed in remote and desert regions, thus expanding the aerosol observation network. Overall, this is a well written paper with good scientific merit. I only have a few minor questions.

**Response:** Thank you for giving us the opportunity to improve the quality of this manuscript. We have substantially revised this manuscript by following your insightful comments and constructive suggestions. Please find out our point-by-point responses below. We have studied comments carefully and have made correction which we hope meet with approval. Revised portion are marked in **red** in the revised paper.

Minor comments:

1. The design of CW193 is very similar to that of CE318. The authors indicated that the biggest advantage is CW193's portability. I suggest providing more detailed description about this. In Figure 2, only the optical head part is shown, or is this the whole system? If latter, I suggest making it clear as this indeed appears much more compact than CE318.

**Response:** Thanks for your constructive comment. Yes, the whole device is shown at the left part in Figure 2, which is consist of optical head, robotic drive platform and stents system. These three parts can be easily connected together only by a few screws. As for the its portability, it is really a main character for CW193. Except for its highly integrated design, the cross weight is about 12 kg, and this make it easier to transport. We have rewritten corresponding sentences in our paper.

*Lines 143-147 in the revised paper:*
*"The instrument is mainly composed of three parts: optical head, robotic drive platform, and stents system (as shown in the left part of Figure 2). These three parts can be easily connected together only by a few screws. Except for its highly integrated design, the cross weight of CW193 is about 12 kg, and this make it easier to transport. Specifically, we presented the comparison of technical specifications between CE318-N and CW193 in table 1."*

2. Does the retrieval of aerosol optical properties use the same inversion method as AERONET? Please briefly describe the retrieval method.

**Response:** Thanks for your kind suggestion. We used similar inversion method as AEROENT, which is developed by Dubovik et al. (2002, 2006) and (García et al., 2008, 2012). In our revised paper, we have re-organized the section 2.2.3 to present the data processing method in this campaign, including AOD, WV, VSD, SSA, ADRF and their uncertainties.

*Lines 220-229 in revised paper:*
*" As for the inversions of VSD and SSA in this campaign, they were retrieved from the observational data from the diffuse-sky measurements of the CW193 at 440, 670, 870, and 1020 nm using the*

*algorithms of Dubovik et al. (2002, 2006). The ADRF was calculated by the radiative transfer module, which is similar to the inversion of AERONET (García et al., 2008, 2012). Because the introduction, validation and application of these inversions and their algorithms have been presented in many previous studies based on CARSNENT observation, we did not repeat these again in this paper (Che et al., 2018, 2019c; Zhao et al., 2018; Zheng et al., 2021). In general, the AODs' uncertainty was 0.01 to 0.02 (Eck et al., 1999). The VSD accuracy was 15 % to 25 % between 0.1 µm ≤ r ≤ 7.0 µm while 25 % to 100 % for other radius (Dubovik et al., 2002). The SSA accuracy was 0.03 when its was calculated under the condition of AOD440 nm >0.50 with a solar zenith angle >50 ° (Dubovik et al., 2002). The bias for measured radiation at the surface was about 9±12 W m$^{-2}$, affected by the dominant aerosol type (García et al., 2008)."*

3. In addition to comparing with AERONET and CARSNET, I think it is also very important to independent evaluate the measurement and retrieval accuracies of CW193. How accuracy are the sky and diffuse radiances? How are the errors in these measurements transferred to the retrieved products? Are these accuracy levels comparable, or better than AERONET?

**Response:** Thanks for your constructive suggestion. Yes, the independent validation of measurement and retrieval accuracies is an important processing for the instrument evaluation. As for the measurement, the wildly used mothed is the inter-comparison based on ground-based observation of broadband fluxes. In next step, we plan to conduct this inter-comparison at the radiation calibration center of Chinese Academy of Sciences in Dunhuang (40.15°N, 94.69°E, 1140 m a.s.l in Northwest China). So we just showed the corresponding results of aerosol microphysical, optical, and radiative properties in the present study. As for the uncertainties in retrieval method, we have added the data processing method, including AOD, WV, VSD, SSA, ADRF and their uncertainties in the re-organized section 2.2.3, as have mentioned above. As the result, considering the performance of these products, we concluded that the CW193's inversions are comparable with the AERONET.

*Lines 220-229 in revised paper:*
*"As for the inversions of VSD and SSA in this campaign, they were retrieved from the observational data from the diffuse-sky measurements of the CW193 at 440, 670, 870, and 1020 nm using the algorithms of Dubovik et al. (2002, 2006). The ADRF was calculated by the radiative transfer module, which is similar to the inversion of AERONET (García et al., 2008, 2012). Because the introduction, validation and application of these inversions and their algorithms have been presented in many previous studies based on CARSNENT observation, we did not repeat these again in this paper (Che et al., 2018, 2019c; Zhao et al., 2018; Zheng et al., 2021). In general, the AODs' uncertainty was 0.01 to 0.02 (Eck et al., 1999). The VSD accuracy was 15 % to 25 % between 0.1 µm ≤ r ≤ 7.0 µm while 25 % to 100 % for other radius (Dubovik et al., 2002). The SSA accuracy was 0.03 when its was calculated under the condition of AOD440 nm >0.50 with a solar zenith angle >50 ° (Dubovik et al., 2002). The bias for measured radiation at the surface was about 9±12 W m$^{-2}$, affected by the dominant aerosol type (García et al., 2008)."*

4. Could the authors provide some explanations of the differences between CW193 and

AERONET/CARSNET? Based on Figures 7-11, there are still some biases and differences.

**Response:** Thanks for your suggestions. As for the AOD from direct Sun radiance measurement, the main calculated method is based on Beer's law, in which the total extinction is mainly affected by aerosol extinction, water extinction, Rayleigh scattering and gas absorption (e.g., $NO_2$, $O_3$). Considering these variables and the assumption in the algorithm, the total AODs' uncertainty was 0.01 to 0.02 according to Eck et al. (1999). In this campaign, the calibration coefficient of CW193 was transferred from the master instrument of AERONET by inter-comparison. As reported by Che et al. (2009), the differences in coefficient transfer at 440, 675, 870, 1020 nm were about 2% between CARSNET and AERONET. In this study, the AOD bias was mostly concentrated within ±0.04 (4%), so we concluded the results of AODs were accurate with acceptable difference. For VSD, SSA and ADRF, these retrievals uncertainties, in fact, are greatly affected by the calibration processing, because there is no absolute self-calibration procedure between the sphere calibration, indicating the differences of retrievals were joint determined by many factors, such as uncertainties of inherent algorithm assumption, input direct and sky radiance, surface albedo. In order to reduce the uncertainty from input radiance, we only used the results, which the observation interval is within 10 minutes to conduct the comparison (Lines 427-436). Through there are still some differences, we suggested that these results were comparable with the AERONET. In next step, we will further test its stability and accuracy based on long-term observation campaign, with the reference of AERONET results.

---

## Author Comment (AC3)

**Referee #3**

General comments:

Atmospheric aerosols have significant influence on regional air quality, regional climate change, as well as human health. Their loadings have been increased substantially compared with those in Pre-industrial times. A detailed description on the aerosol optical and physical properties is the prerequisites for better evaluating the effects of the aerosols. Unfortunately, uncertainties of the aerosol radiative forcing and climate effects still exist due to a lack of knowledge about the aerosol properties. Therefore, a new highly integrated observation instrument is necessary to be developed to fill the gap of current observation system. This study proposes a new multispectral photometer (CW193) with a highly integrated designing and smart control performance for monitoring aerosol microphysical, optical, and radiative properties. The results indicate that CW193 can well observe and capture the aerosol characteristics by comparing with AERONET products, implying that the instrument may have a wide application prospect in the further. The topic of this study is interesting and novel. Therefore, the paper has a potential for publication in the journal.

**Response:** Thank you for giving us the opportunity to improve the quality of this manuscript. We have substantially revised this manuscript by following your insightful comments and constructive suggestions. Please find out our point-by-point responses below. We have studied comments carefully and have made correction which we hope meet with approval. Revised portion are marked in **red** in the revised paper.

Specific comments:

1. Why the new instrument is named as CW193? The authors can make a detailed introduction.

**Response:** Thanks for your constructive suggestion. Yes, the name of CW193 contains lots of meanings. Firstly, the CW means the **C**hinese device for **W**orld. We hope that this instrument could meet the international standard for aerosol monitoring. Also, the it represented the inter-comparison in this paper was conduct and affiliated in the **C**AMS atmospheric composition **W**atching program. Last but not least, we hope the CW193 to show respect to CE-318, as the latter is the wildly used device in the world with high accuracy and stability—that is "**C**E-318's quality is **W**anted". However, in order to make our paper more concise, we decided only to show the first meaning of "**C**hinese device for **W**orld" after the discussion with all author, and we think this point could express our quality requirement, confidence and best wishes to CW193.

2. What is the main difference (or progressiveness) of the CW193 against to the CE-318?

**Response:** Thanks for your comment. We suggested that the main difference of CW193 against CE-318 is CW193's portability (highly integrated design). As the left part in Figure 2 shows, the whole device is consisted of optical head, robotic drive platform and stents system. These three parts can be easily connected together only by a few screws. Except for its highly integrated design, the cross weight is about 12 kg, and this make it easier to transport. We have rewritten corresponding sentences in our paper as follow to highlight this difference.

*Lines 143-147 in the revised paper:*

*"The instrument is mainly composed of three parts: optical head, robotic drive platform, and stents system (as shown in the left part of Figure 2). These three parts can be easily connected together only by a few screws. Except for its highly integrated design, the cross weight of CW193 is about 12 kg, and this make it easier to transport. Specifically, we presented the comparison of technical specifications between CE318-N and CW193 in table 1."*

3. How many observation intervals can be set for CW193?

**Response:** Thanks for your kind suggestion. In Table 2, we present the observation frequency for Sun measurement of CW193, and measurement is conduct in every 3 minutes, which can be set up to 2 minutes. As for the routine of sky radiance observation (ALM, PPL), the CW193 can conduct continuous observation once the corresponding observation schedule is set. We have added these supplementary notes in the Table 2 as follow.

*Line 202 in revised paper:*
*"*

| Observation frequency for sky radiance | ... | ... |
|---|---|---|
| Observation schedule** | SUN, ALM, PPL | - SUN, ALM, PPL (in default)
- only SUN (Optional)
- only ALM (Optional, **consecutive**)
- only PPL (Optional, **consecutive**) |
| Monitoring Software | ... | ... |

*"*

4. The authors state that CW193 has a low maintenance requirement. How long and in what conditions does it need to be taken to maintain? I think all the ground-based instruments are needed to have a routine maintenance.

**Response:** Thanks for your constructive suggestion. We all agree that the routine maintenance is an important and necessary process in the observation campaign. At line 19 in the Abstract, this misleading sentence have been corrected. We intended to state that the CW193 is appropriate for the deployment in remote and unpopulated regions due to its portability, including highly integrated design and smart control performance. So we used "these characteristics" to represent the difference against CE318 in our revised paper as follow.

*Lines 19-20 in revised paper:*
*"Because of these characteristics, this instrument is appropriate for the deployment in remote and unpopulated regions."*

5. To make the instrument more reliable, more observation and validation works should be carried out in the further. For example, the authors can perform a series observation activity with different pollution levels, in different time scales, in different regions as well as in different seasons.

**Response:** Thanks for your constructive comment. We could not agree more that the new device should be tested in detail under different pollution levels, in different time scales, in different regions as well as in different seasons (as shown in lines 596 to 599). As for the verification under the different pollution levels, we have preliminarily tested its performance in this paper with the $PM_{2.5}$ varying from 6 to 104 μg m$^{-3}$. However, its accuracy and stability under heavy pollution is still need to be further assessed when $PM_{2.5}$ exceed 150 μg m$^{-3}$. As for the observation in different seasons and regions, we plan to conduct long-term field campaign in next year considering the various restriction by COVID-19 in this year. It should be note that we found that the WV bias (within ±0.04 in this paper) showed increasing trend with the values in this campaign, which means the performance of CW193 is still need to be further tested in humid summer days (as shown in lines 554-555).

6. Conclusion should be more refined instead of repeating the results. An additional discussion on the potential application of the instrument in the future can be involved in this section.

**Response:** Thanks for your suggestion. We suggested that the CW193's portability (highly integrated design) makes it more appropriate for the deployment in remote and unpopulated regions, to complement the observation gaps of CARSNET. In addition, the optional observation schedule (such as only ALM) could meet the different requirement of the aerosol microphysical, optical, and radiative properties. Especially when the VSD and SSA is in great demand for the modification of numerical model and the verification of satellite inversion products, in only ALM mode, these inversions could be obtained about 2 to 3 times in an hour, while for once in default observation schedule. We have added this discussion and application in our revised paper as follow.

*Lines 606-609 in revised paper:*
*"Due to its smart control performance and optional observation schedule, such as ALM mode, the CW193 could meet the different requirement of the aerosol microphysical, optical, and radiative properties. When the VSD and SSA is in great demand for the modification of numerical model and the verification of satellite inversion products, these inversions could be obtained about 2 to 3 times in an hour, while for once in default observation schedule."*

7. English should be corrected throughout the whole manuscript.

**Response:** Thanks for your kind suggestion. The major change is that we re-organized the section 2.2.3 to present the calibration and data processing in this campaign. Except for this, we have check through this paper and revised some spelling error and grammar mistakes. Also, we revised some phrases and supplementary notes to make our manuscript more logical and concise. Here we presented some minor corrections as follows.

*Line 19-20 in revised paper:*
*"...is composed of three parts (**optical head**, **robotic drive platform**, and **stents system**)."*

*Line 143 in revised paper:*
*"AOD and other retrievals (**such as** microphysical, optical, and radiative properties of aerosols)*

*from Sun radiation…"*

*Line 210-211 in revised paper:*
*"…using the method of coefficient transfer* **(inter-comparison)** *with the reference master instruments of AERONET…"*

*Line 577-578 in revised paper:*
*"**As a result**, the CW193 retrievals in this study showed high precision for SSA and ADRF…"*

---

## Author Comment (AC4)

**Community Comment #1**

Response: The authors are very grateful for your interest and quick comment to our work. Please find out our point-by-point responses below. Revised portion are marked in **red** in the revised paper.

First of all, we should explain the main target of this work here to avoid misunderstanding—to evaluate the performance of aerosol microphysical, optical and radiative properties measured from a multiwavelength photometer, rather than a commercial statements or competition for a monitoring instrument. In this campaign, we serve as the observation and data processing platform to conduct this intercomparison work with the reference of AERONET's result at "Beijing-CAMS". With that in mind, we intend to change the title of this paper from "A new multispectral photometer for monitoring aerosol microphysical, optical, and radiative properties" to "Evaluation of aerosol microphysical, optical and radiative properties measured from a multiwavelength photometer", to emphasize the principal target of this intercomparsion work. Also, aiming at this topic, some statements of little relevance will be revised in our paper, such as cost and weight. Therefore, this assessment work will contribute to the scientific research such as aerosol measurement, but **not for the commercial purpose nor for the instrument competition**.

Secondly, the scientific meaning of this work should be restated here to make it more clearly. As reported by WMO-GAW's report No. 207, 227 and 228 (2012; 2016; 2017), the multiwavelength aerosol optical depth (AOD) is still recommended as the long-term measurement variables at the GAW's implementation plan from 2016 to 2023. Particularly via ground based AOD attenuation observation, it is regarded as the highly accurate monitoring method to provide indispensable data for satellites validation and global modelling. Additionally, according to WMO's guideline, an absolute limit to the estimated uncertainty of 0.02 optical depths for acceptable data and <0.01 as a goal to be achieved in the near future. This guideline highlighted that data assessment is as important as the data observation. For this sake, we suggested that the work in this paper has practical significance. On the one hand, we tested the accuracy of multiwavelength AOD under various environmental conditions, including low and high aerosol loading (different PM concentration levels), clear and cloudy days (cloud contaminated). On the other hand, the retrievals evaluation is also provided such as single scattering albedo (SSA), volume size distribution (VSD) and radiative forcing, which are the importing parameters for the climate modelling. Thus, **this work could contribute to obtaining accurate AOD data and reduce its uncertainty in response to GAW's target**.

The last but not the least, the relevant statement of AERONET and its CE318 photometer in this paper is regarded as the criterion reference to test the observation results of CW193, but by no means as the competitor network or instrument. As recommended by Working Group II at WMO-GAW's report No. 162 (2004), the international coordination of AOD networks is inadequate and could be improved by a

federated network under the WMO/GAW umbrella, and networks should become traceable and maintainable via intercomparisons and calibrations. We all know that the AERONET is the most widely network around the world, which is mainly is composed of CE318 photometer, to provide quality assured aerosol optical products. Up to now, there are many photometers except CE318 and CW193 have realized the function of AOD measurement in China, such as DTF-5 and PSR-2 (Li et al., 2012; Huang et al., 2019). However, we suggest here also in our paper, all the instrument and its products should meet the WMO/GAW's criterion and keep consistency with AERONET, providing comprehensive, comparable aerosol optical products. Owing to the above factors, we selected the results of "Beijing-CAMS" in AERONET as the reference to assess the data of CW193. On the one hand, the CE318s, five master instruments, are periodically calibrated at Izaña observatory in every six months, indicating the instruments and their calibration coefficients are reliable enough for field calibration via intercomparison. On the other hand, the similar retrieval algorithm (Dubovik et al, 2002; 2006) with AERONET had been tested by previous studies based on CARSNET and can reduce the inversion bias as much as possible, though these biases may be affected by various factors such as sphere calibrations uncertainty. For these reasons above, we conduct this campaign **to present an overall assessment of AOD accuracy and inversion comparability with the reference of AEROENT, aiming at keeping consistency with AERONET rather than to replace it**.

In summary, we are much obliged to your community comments for pointing out the non-standard statements in this evaluation work. We will substantially revise this manuscript by following your insightful comments and constructive suggestions. As an observation and evaluation platform, we hope more instruments will appear to meet the WMO/GAW's criterion or AERONET' accuracy, which will be a great assistance to combat climate change.

References

Dubovik, O., Holben, B., Eck, T. F., Smirnov, A., Kaufman, Y. J., King, M. D., Tanré, D. and Slutsker, I.: Variability of Absorption and Optical Properties of Key Aerosol Types Observed in Worldwide Locations, J. Atmos. Sci., 59(3), 590–608, doi:10.1175/1520-0469(2002)059, 2002.

Dubovik, O., Sinyuk, A., Lapyonok, T., Holben, B. N., Mishchenko, M., Yang, P., Eck, T. F., Volten, H., Muñoz, O., Veihelmann, B., van der Zande, W. J., Leon, J. F., Sorokin, M. and Slutsker, I.: Application of spheroid models to account for aerosol particle nonsphericity in remote sensing of desert dust, J. Geophys. Res. Atmos., 111(D11), 11208, doi:10.1029/2005JD006619, 2006.

Li, J., Jia, L., Xu, W., and Wei, H. Comparison Certification and Error Analysis of Atmospheric Optical Parameters Measured by DTF Sun-Photometer, Journal of Atmospheric and Environmental Optics, 7(2), DOI: 10.3969/j.issn.1673-6141.2012.02.002, 2012. (In Chinese)

Huang, D., Li, X., Zhang, Y. and Zhang, Q. Novel high-precision full autocontrol multi-waveband sun photometer, Journal of Applied Optics, 40(1), DOI:

10.5768/JAO201940.0105001, 2019. (In Chinese)

U. Baltensperger, L. Barrie and C. Wehrli. Geneva, WMO/GAW experts workshop on a global surface-based network for long term observations of column aerosol optical properties[J] World Meteorological Organization, 2004. (GAW Report No. 162)

Lund Myhre C, Baltensperger U, Barrie L, et al. Recommendations for a composite surface-based aerosol network[J]. World Meteorological Organization, 2012. (GAW Report No. 207)

Zhongming Z, Linong L, Wangqiang Z, et al. WMO/GAW Aerosol Measurement Procedures, Guidelines and Recommendations[J]. World Meteorological Organization, 2016. (GAW Report No. 227)

World Meteorological Organization. WMO Global Atmosphere Watch (GAW) Implementation Plan: 2016－2023[J]. World Meteorological Organization, 2017. (GAW Report No. 228)

**A. General comments**

The manuscript presents a system for the monitoring of atmospheric aerosols, based on a new instrument. It is mainly based on a comparison with the AERONET system (based on the CE318 photometer), considered as the reference.

The main claims of the authors are

- novelty,
- additional functions,
- validity of metrology,
- validity of data processing chain,
- validity for network operations,
- simplified maintenance,
- low cost.

Having red this paper, I concluded that these claims are not substantiated by scientific and technical evidences:

1. No conceptual novelty is shown hence the work appears rather like an approximate duplication of the whole AERONET instrument and system.

**Response:** Thanks for your comments. The principal goal of this paper is to evaluate the performance of aerosol microphysical, optical and radiative properties measured from a multiwavelength photometer with the reference of AERONET, rather than a commercial statements or competition for a monitoring instrument. We have restated this point at the above discussion and will revise the corresponding sentences in our paper.

2. The comparison of CW193 specifications to the reference CE318 system is not complete and not fair:
   a) First, this comparison should be done with the current AERONET reference instrument CE318T and not with the old version of CE318 as done in the paper

**Response:** Thanks. This paper is aiming at presenting data evaluation rather than the instruments competition. We chose main aerosol optical products of CW193 to conduct this assessment, and all the reference data were downloaded from the AERONET website. Since the CE318 is the main instruments that AEROENT used and the present version of CW193 could obtain data from direct Sun measurement and almucantar scan, we revised the corresponding tables with the reference of CE318-T mode. Anyway, no matter what kind of instrument it is, we suggested that data accuracy should meet the WMO/GAW's guideline and be comparable.

b) Second, and linked to point a), several important functions present in AERONET are lacking (Lunar measurements, polarized sky radiance option, multiple scenario configurations). Hence, this comparison looks unfair.

**Response:** Thanks. This comparison focused on data accuracy and comparability, rather than the monitoring network. We will revise some statement with small-relevance under this topic. Actually, for the present version of CW193, the Lunar measurements and polarized sky radiance is not available.

c) Third, the claimed benefits of some new features brought by CW193 are not explained nor proven.

1. CW193 performances are not characterized nor validated
2. Long term performance including robustness, sensitivity to weather conditions is not evaluated, therefore not validated.
3. In the paper, the data quality analysis is limited to a few selected measurement days. The evaluation of the system's quality requires a much more comprehensive experimental plan.
4. The additional benefits claimed for the improvement of operational observations (robustness, simple maintenance, low cost) are not evidenced.

**Response:** Thanks. We mainly compared the data accuracy and comparability with the reference of AEROENT. As a scientific paper, we considered that the introduction of basic design and parameters for an instrument is acceptable, because the target is data evaluation rather that instrument comparison. As for the system's quality, we agree that the long-terms observation is need. So far, the CW193 have been running at city Changchun (at the Northeast China, 125.35°E, 43.88°N) from 2020 to 2021 for the low temperature test at winter (~ -30°C of minimum temperature) as below shows. However, to evaluate its AOD and inversions' accuracy and comparability, the criteria is important since the AOD bias <0.02 is acceptable according to WMO/GAW's suggestion. So we conducted this synchronous observation at "Beijing-CAMS", considering the AEROENT results could be a standard, aiming at keeping consistency with AERONET. Additionally, after the discussion, we think this paper should be more focused on data evaluation and some statements that may cause misunderstanding will be modified, such as simple maintenance, low cost.

[Figure]

Figure 1. Examples of Raw digital signals at Changchun under the temperature range from ~ -30 to 35 °C.

As conclusion,

- The work presented in the paper does not bring new knowledge to the scientific community, as it would expect.
- It mainly rather makes a series of technical and commercial statements on claimed advantages of CW193 instrument without providing corresponding evidences.
- In summary, these weaknesses and lacks are in opposition/contradiction with the claimed advantages of CW193: *novelty, additional functions, validity of metrology, validity for network operations, simplified maintenance, low cost.*

**Response:** Thanks for your constructive comments. Actually, we suggested that data assessment is as important as the data observation to meet the WMO/GAW's criterion or AERONET' accuracy, which will be a great assistance to combat climate change. Second, the introduction of the instrument is only used as an auxiliary description in this paper, rather than main topic. Also, we will modify some statements that may cause misunderstanding such as simplified maintenance, low cost as the response to specific comments below.

**B. Detailed and specific comments**

1. Line 83: "wired communication (for example, serial communication via RS-232) between the instruments and a personal computer is still necessary for most CE318-N photometers".

*Comment: This statement is NOT correct. AERONET operates a large number of sites at remote locations without wired communication with a PC.*
**Response:** Thanks. We are not aiming at the network or instruments comparison. This part has been deleted in our revised paper.

2. Line 85: "the non-integrated instrument components, such as the control unit, external battery, protection box, and stents platform, not only cause most of the operational problems but also make the deployment and maintenance difficult for staff with inadequate training"

*Comment: This opinion is not justified and does not seem fair. In AERONET, the protection box and simple tripod platform are options that may be very useful for some types of installations, especially in remote places, where trained staff and technical means are not available. The modular design of CE318 is often an advantage in terms of easy replacement of parts.*
**Response:** Thanks. We are not aiming at the network or instruments comparison. This part has been deleted in our revised paper.

3. Lines 92-93: "which makes the whole system efficient, secure, low cost and highly integrated."

*Comment: this list of assertions is not justified by the information provided in the paper. The presented integrated design does not allow local control of the instrument without a PC, which may be a major issue in remote locations. The low cost should be quantified, including initial and expected maintenance costs, and spread over the proven expected lifetime of the instrument. Efficiency and security should be quantified over the long term, in terms of uptime of the instrument and proportion of data brought to some defined quality level. AERONET has proven an unmatched efficiency and service level in producing quality assured atmospheric aerosol products over the long term*
**Response:** Thanks. We are not aiming at the network or instruments comparison. This sentence has been revised as below.

*Lines 109-110 in the revised paper:*
*"...which makes the whole system efficient and highly integrated."*

4. Lines 97-99: these assertions are not justified.

**Response:** Thanks. We are not aiming at the network or instruments comparison. This sentence has been revised as below.

*Lines 97-99 in the revised paper:*
*"These features make the CW193 a particularly suitable multiwavelength photometer for monitoring aerosol microphysical, optical, and radiative properties, which is contribute to verifying the satellite and modelling products"*

5. Line 125: "largest" should be qualified: probably refers to China

**Response:** Thanks. It has been qualified according to your suggestion as below.

*Lines 97-99 in the revised paper:*
*"These features make the CW193 a particularly suitable multiwavelength photometer for monitoring aerosol microphysical, optical, and radiative properties, which is contribute to verifying the satellite and modelling products"*

6. Line 126: "Same algorithm" *should be qualified. How has it been validated?*

**Response:** Thanks. It has been revised according to your suggestion as below.

*Lines 128 in the revised paper:*
*"CARSNET uses the similar algorithm as AERONET…"*

7. Line 143: *Again, comparison with CE318-N is not relevant as this is an old version of CE318. Most of AERONET sites are equipped with the more recent version CE318-**T**. The table should be corrected to present a fair comparison.*

**Response:** Thanks. We are not aiming at the network or instruments comparison. Because the AERONET use CE318s as the master and field instruments, we modified the table 1 with the reference of CE318-T mode. As a result, in this table, some of the parameters that of small-relevant with the main topic have been deleted such as weight, dimensions. Please find the revised one below.

*Lines 150 in the revised paper:*
*"Table 1. Technical specifications for CE318-T* and CW193*

| | CE318-T | CW193 |
|---|---|---|
| *Main components* | *Optical head, Control unit, Robot,* | *Optical head, Robotic drive platform, Stents system* |

| | | |
|---|---|---|
| *Spectral range* | *340, 380, 440, 500, 675, 870, 937,1020, 1640 nm* | *340, 380, 440, 500, 675, 870, 937,1020, 1640 nm* |
| *Field of view* | *1.26°* | *1.30°* |
| *Detection's azimuth range* | *0° to 360°* | *0° to 360°* |
| *Detection's zenith range* | *0° to 180°* | *0° to 180°* |
| *Sun tracking accuracy* | *0.01°* | *0.02°* |
| *Communication outputs* | *RS232, USB, UMTS/3G/W-CDMA, GPRS* | *RS232, 4G* |
| *Storage* | *Flash memory (4 MB), SD card (32 G)* | *SD card (32 GB)* |
| *Power supply* | *Power adapter (110 to 240 V), Solar panel (5 W), External batteries (12 V, 16Ah)* | *Power adapter (110 to 240 V), Solar panel (30 W)* |
| *Software* | *PhotoGetData* | *DataMonitor* |

\*Photometer for CE318-T mode in standard version"

8. Table 1 - *The whole table 1 should be corrected with CE318-T technical specifications*

**Response:** Thanks. We have revised the table 1 and please see the response above.

9. Table 1: *what is the type of detector used?*

**Response:** Thanks. This item had been deleted according to the discussion above.

10. Table 1 - Drift of single band filter's transmission rate < 1% for CW193 :
    - Sun tracking accuracy: 0.02 °
    - Temperature range: -30° to 60°
 *The characterization of these performances should be described.*

**Response:** Thanks. In the revised table 1 above, we deleted the temperature, humidity range and drifts.

11. Table 1: - Power supply for CW193
 *The type and capacity of the battery system should be described. The autonomy of the system, in case of operation on the solar generator and absence of direct sun, should be stated.*

**Response:** Thanks for the constructive comments. In table 1, we just listed the standard configurations of power supply.

12. Table 1: - Gross weight and flycase dimensions are not really relevant, or should be completed with net weight and dimensions, and with all components (solar panel)

**Response:** Thanks. Please find the revised table 1 above. We have deleted this item.

13. Lines 174-175: "the design of CW193 is very robust, ensuring long-term steady operation in a wide range of temperature and humidity, between about −30°C and 60°C and between about 0 and 100%, respectively"

*Comment: This assertion is not supported by any evidence.*

**Response:** Thanks for pointing out. The CW193 have been running at city Changchun (at the Northeast China, 125.35°E, 43.88°N) from 2020 to 2021 for the low temperature test at winter (~ -30°C of minimum temperature). Also, we conduct several field observations campaigns at remote stie. Here we present some of details at these campaigns below.

[Figure]

Figure 2. Field observation campaign at Wuhai, Alashan and Dunhuang.

[Figure]

Figure 3. AOD in level 1.0 and level 1.5 during field observation campaigns at Wuhai, Alashan and Dunhuang.

14. Lines 189-192: "It is very convenient to receiving data via 4G network when the serial communication is unavailable in some remote regions, and also in this mode, multiple device control is achievable (device 003, 005 and 006 are on-line and controllable in Figure 3). In the data download part, the history data can be easy downloaded by selecting the start and end time via drop-down menu".

*Comment: This is presented as a new function and an advantage, but the AERONET network already operates a large number of remote sites with direct telecommunication link. In AERONET, full data collection is ensured fully automatically in real time, or even after interruption of communications. This is more convenient than manual control through a software.*

**Response:** Thanks. We just show the basic introduction of the data monitoring software in here rather than the comparison with AERONET's system. We all agree that the AERONET and its system is highly efficient and provides useful data for the global climate.

15. Table 2 - *The whole table 2 should be corrected with CE318-T functional specifications*

**Response:** Thanks. We modified the table 2 according to your constructive suggestion.

*Lines 222 in the revised paper:*
*"Table 2. Functional specifications for CE318-T\* and CW193*

| | CE318-T | CW193 |
|---|---|---|
| *Observation frequency for sun measurement* | *15 mins (in default), up to 2 mins* | *3 mins (in default), up to 2 mins* |
| *Mode of sun tracking* | *At the beginning of every measurement* | *Keep tracking* |
| *Observation frequency for ALM scan* | *According to air mass, when air mass =1.7, 2.0, 2.2, 2.4, 2.6...* | *Every integral local time at 7, 8, 9,10, 11...19 O'clock (primary)*
 *According to air mass, when air mass =1.7, 2.0, 2.2, 2.4, 2.6... (subsidiary)* |
| *Observation schedule\*\** | *Sun, Moon, Black, Principal plane, Almucantar, Hybrid, Cross Sun, Cross Moon. Curvature Cross.* | *- Sun, Black, Almucantar, Principal plane (in default)*
 *- only Sun (optional)*
 *- only Almucantar (optional, consecutive)*
 *- only Principal plane (optional, consecutive)* |
| *Monitoring Software* | *- instruments setup*
 *- wavelengths selection*
 *- scan modes & scenarios* | *- scan modes & scenarios configuration*
 *- measurement scheduling* |

| configuration | - wavelengths selection |
| - measurement scheduling | - data visualization |
| - data analysis | - data retrieval |
| - data visualization | - data storage (TXT files) |
| - data storage (raw data, k8, ASCII files) | - commands inputs |
| | - multidevice control (4G mode only) |

*Photometer for CE318-T mode in standard version
**Photometer in auto mode"

16. Line 211: "five instruments"

*Comment: The method and specifics of the calibration of the studied CW193 should be described. For the whole intercomparison study, Sun calibration should be made on a different set of data. Is it the case?*

**Response:** Thank you so much for your constructive comments. In this campaign, the CW193 (could be regarded as field instrument) was calibrated via coefficient transfer method (inter-comparison) with the reference of AERONET master instruments according to the Eq. 1 as below,

$$C(\lambda) = C(\lambda)_0 \times \left( \frac{V(\lambda)}{V(\lambda)_0} \right) \ldots \ldots Eq.\,1$$

where the $C(\lambda)$ and $C(\lambda)_0$ is the calibration coefficient for field instrument and master instrument at $\lambda$ wavelength, respectively. $V(\lambda)$ and $V(\lambda)_0$ is the digital count for field instrument and master instrument at $\lambda$ wavelength, respectively. We have rewritten this sentence in paper and added one corresponding reference of coefficient transfer method to make it more accurate.

*Lines 204-205 in revised paper:*
*"...using the method of coefficient transfer (inter-comparison) with the reference master instruments of AERONET (Che et al., 2009, 2019c; Zheng et al., 2021)."*

*Che, H., Zhang, X., Chen, H., Damiri, B., Goloub, P., Li, Z., Zhang, X., Wei, Y., Zhou, H., Dong, F., Li, D. and Zhou, T.: Instrument calibration and aerosol optical depth validation of the China Aerosol Remote Sensing Network, J. Geophys. Res. Atmos., 114(D3), doi:10.1029/2008JD011030, 2009.*

17. Line 313-314: "Therefore, in summary, the CW193 shows high stability under both high and low aerosol loadings; hence, the excellent detection ability makes it a reliable instrument for aerosol monitoring."

*Comment: This conclusion regarding the reliability of the instrument is not justified in the paper.*

**Response:** Thank you so much for your constructive comments. In this part, we intended to show the wavelength dependence of AOD measured from CW193 both at

clear and polluted days. We have revised these sentences according to your suggestion as below.

*Lines 204-205 in revised paper:*
*"...the CW193 showed good ability of AOD's wavelength dependence under both high and low aerosol loadings..."*

18. Figure 6:
*Comment: This figure does not show interruption at nighttime. The observation time per day should be explained.*
**Response:** Thanks for your constructive comments. At present version of CW193, the Lunar observation is not available as the revised table 2 and discussions show above. We have revised this sentence according to your suggestion as below.

*Lines 319 in revised paper:*
*"Figure 6 shows the diurnal variation of cloud-screened AOD (only from daytime observation) for each band from CW193 during this campaign"*

19. Line 352: "We set the envelopes as ±(0.05 + 10%)."
*Comment: The choice of this criterium should be explained. It is quite large compared to the AERONET uncertainty.*
**Response:** Thanks for your constructive suggestion. Actually, this envelope is widely used in many previous studies aiming at AOD validation from satellite, to show the error range versus reference. We used this figure here to highlight the AOD precision with statistical parameters, as well as the accuracy of ground-based observation against satellite monitoring. We all agree that the AERONET provide the high accuracy AOD data with uncertainty smaller than 0.02. And the specific analysis of AOD bias can be found in Figure 8 in our revised paper.

20. Line 412: "uncertainty of <10% is acceptable for the discussion"
*Comment: This level of uncertainty is much higher than AERONET's.*
**Response:** Thanks for your suggestions. This statement is inaccurate and we have deleted it in our revised paper.

21. Lines 568-570
*Comment: This conclusion should be expressed as a preliminary only. It must be checked on long-term and various weather and aerosol conditions.*
**Response:** Thanks for your constructive suggestion. Yes, we all agree that the stability of CW193 must be checked on long-term and various weather and aerosol conditions. So we just present the preliminary evaluation of these data with the reference of

AERONET in this study, showing its consistency with AERONET. We have revised this sentence according to your suggestion as below.

*Lines 591 in revised paper:*
*"The results of this preliminary evaluation indicate that…"*

22. Lines 582-585: "the highly integrated design and smart control performance make CW193 more convenient and suitable for the aerosol monitoring, providing similar aerosol optical properties to AERONET. In addition, owing to the built-in 4G communication module, CW193 could be used to create networks in an inexpensive and simple way."

*Comment: as such, this is a commercial statement, not evidenced by the paper. It should be removed or rephrased.*

**Response:** Thanks. We have rephrased this part according to your constructive suggestion as below.

*Lines 604 in revised paper:*
*"Above all, the highly integrated design and smart control performance make CW193 suitable for the monitoring microphysical, optical, and radiative properties of aerosol. Due to its smart control performance and optional observation schedule, such as ALM mode, the CW193 could meet the different requirement of the aerosol microphysical, optical, and radiative properties. When the VSD and SSA is in great demand for the modification of numerical model and the verification of satellite inversion products, these inversions could be obtained about 2 to 3 times in an hour, while for once in default observation schedule. As a result, this instrument could be regarded as a contributor in regional and climate model data assimilation, satellite modification, and improving knowledge of the temporal and spatial variations of aerosols."*

---

## Referee Report (RR1)

**Evaluation of aerosol microphysical, optical and radiative properties measured from a multiwavelength photometer**

**By Zheng et al.**

**General comments:**

The authors have well addressed the issues of concern to the reviewers. And the revised manuscript has a substantial improvement. Therefore, I recommend that this manuscript can be published in AMT.

---

## Referee Report (RR2)

This paper presented an evaluation of aerosol microphysical, optical and radiative properties measured from a multiwavelength photometer, named CW193. As introduced by the authors, the instrument has a highly integrated design, smart control performance, and is composed of three parts (optical head, robotic drive platform, and stents system). Then the CW193 product was inter-compared and validated using reference data from the AERONET based on the synchronous measurements. The results of this preliminary evaluation indicated that the CW193 is appropriate for monitoring aerosol microphysical, optical, and radiative properties, characterized by the good agreement of raw digital counts, accurate AOD results and comparable retrievals with AERONET. In summary, this paper is a good work and has lots of general interest for Atmospheric Measurement Techniques and related communities. Therefore, I have no more major comments and have recommended for acceptance after a minor revision. I suggested the following few comments may improve and strengthen the quality of the manuscript.

**Specific Comments:**

Line no. 61-64: Authors can be mentioned about the limitation of the polar orbiting or lower earth orbiting satellites with relevant references. For example, due to poor spatial and temporal resolution of such satellites, there are about 50% data lost over high-altitude sites mountainous sites in particular at 0.05x0.05 degree spatial resolution of MODIS (Terra) data (Ningombam et al., 2021).

Line no. 79-80: It is also very important to expand such robotic measurement made at high-altitude and mountainous region where there are limited ground based data available due to harsh climatic condition and lack of manpower support for operating the instruments.

Line no. 134: Please put the unit of water vapor (mm or cm ?) after +/- 0.10. Also, I found several places in the manuscript where the authors did not put the unit.

Line no. 249: Please mention which version of AERONET data is used as a reference in the present work.

Line no. 306-308: Authors may be added few more relevant references about the importance of quality controlled data over high-altitude and clean environments where the estimated aerosol parameters are of the order of measurement uncertainties.

Line no. 315: Table 4: $PM_{10}$ for Level I on 7 November is found to be high. Please check if there are any issues in the data. Moreover, aerosol measurement on the same day for Figure 6 might have disrupted due to frequently passing cloudy which may be attributed the high AOD.

Line no. 346: Please correct the wavelength range '70 nm', I think it must be 870 nm.

Figure 12: Please put the unit of water vapor (mm or cm ?) in the Figure. Also, I found several places where the authors did not put the unit.

**References:**

Shantikumar S Ningombam, H-J Song, S K Mugil, Umesh Chandra Dumka, E J L Larson, Brijesh Kumar, Ram Sagar, Evaluation of fractional clear sky over potential astronomical sites, *Monthly Notices of the Royal Astronomical Society*, 2021, Volume 507(3),pp.3745–3760, https://doi.org/10.1093/mnras/stab1971.

---

## Author Response (AR2)

**Referee #4**

This is a very interesting paper presenting a new instrument for measuring aerosol optical properties.
**Response:** Thanks.

I have read the previous reviewer comments and the new manuscript and I could say that most of them has been taken into account.
**Response:** Thank you.

One aspect is that the instrument does a similar "job" with the AERONET/CIMEL instrument. The authors way to prove that the instrument provides good results is the agreement with the CIMEL instrument. So, I think discussion on comparing which one is "better" cannot be included.
**Response:** Thank you for your constructive comment. As we have mentioned in the response to Community Comment #1, the main target is to evaluate the performance of aerosol microphysical, optical and radiative properties measured from a multiwavelength photometer, rather than a commercial statements or competition for a monitoring instrument. We all agree that as an evaluation work, the discussion on comparing which one is "better" should not be included. So, we have check through this paper and substantially revised relevant statements. Revised portion are marked in **red** in the revised paper (tracked changes).

By the way, comparing with the CIMEL and having a good agreement does not mean also an accurate representation of reality. Moreover, as both instruments use the same calibration and same post processing algorithms and assumptions especially in the inversion products.
**Response:** The reviewer's suggestion is really appreciated. We all agree that the good agreement with CIMELs does not mean an absolutely accurate reality, since the they exactly have inherent algorithm and assumptions uncertainties in AOD and other inversions. With that in mind, as well as the practicality of this evaluation work, we used "Beijing-CAMS" 's results from AERONET' as the reference. Because all the devices in this campaign are running at the observation platform of CAMS, the departure originated from spatial difference could be regarded as inexistent, to minimize the errors with real results. In our plan for next steps, we will conduct the long-term validation of CW193 not only in different regions but also with different reference results, such as POM-02 and PFR, to further assess the performance of CW193. We have added this in our revised paper as below.

*Line 601 in revised paper (Line 601 in clean version)*
*"the instruments still need to be further tested under different environment conditions, including long-term observations in mountainous, coastal, and desert regions with the reference of CE318, POM-02 and PFR."*

One aspect that I am missing is a discussion on the temperature dependence of the instrument. AERONET/CIMELs lately have been presenting these corrections and it would be important to understand: how the temperature dependence have been identified and characterized, if it is

considered the same for all instruments and what are the ways of post corrections.

**Response:** Thanks a lot for your constructive suggestion. The temperature response of instrument and its configuration, correction is an essential part for the data evaluation, as it could have certain influence on the calculation for infrared channel measurements. In fact, we also found in this campaign that greater departures in near-infrared bands, which could be influenced by ambient temperature, as show in Line 360 (Line 359 in clean version) in. However, the sensitive experiment of temperature was not presented in this work due to the limited by observation period and temperature chamber. In future, we plan to conduct the sensitive experiment of temperature to determine the temperature coefficients via a larger range of temperature chamber.

A minor comment is in the new text line 105 needs re writing

**Response:** The comment is really appreciated. We have revised corresponding statements according to review's suggestion as below.

*Line 95 in revised paper (Line 94 in clean version)*
*"So far, except for CE318, POM-02 (Nakajima et al., 2020) and PFR (Kazadzis et al., 2018), there are many photometers have realized the function of AOD measurement in China…"*

*Reference*
*Kazadzis, S., Kouremeti, N., Nyeki, S., Gröbner, J. and Wehrli, C.: The World Optical Depth Research and Calibration Center (WORCC) quality assurance and quality control of GAW-PFR AOD measurements, Geosci. Instrumentation, Methods Data Syst., 7(1), 39–53, doi:10.5194/GI-7-39-2018, 2018.*

In general, I think it is an interesting work that it is adequate to the AMT journal standards.

**Response:** Thanks! We all greatly appreciate the reviewer's insightful comments and constructive suggestions for improving the quality of our paper.

**Referee #5**

This paper presented an evaluation of aerosol microphysical, optical and radiative properties measured from a multiwavelength photometer, named CW193. As introduced by the authors, the instrument has a highly integrated design, smart control performance, and is composed of three parts (optical head, robotic drive platform, and stents system). Then the CW193 product was inter-compared and validated using reference data from the AERONET based on the synchronous measurements. The results of this preliminary evaluation indicated that the CW193 is appropriate for monitoring aerosol microphysical, optical, and radiative properties, characterized by the good agreement of raw digital counts, accurate AOD results and comparable retrievals with AERONET. In summary, this paper is a good work and has lots of general interest for Atmospheric Measurement Techniques and related communities. Therefore, I have no more major comments and have recommended for acceptance after a minor revision. I suggested the following few comments may improve and strengthen the quality of the manuscript.

**Response:** Thank you for giving us the opportunity to improve the quality of this manuscript. We have substantially revised this manuscript by following your insightful comments and constructive suggestions. Please find out our point-by-point responses below. We have studied comments carefully and have made correction which we hope meet with approval. Revised portion are marked in **red** in the revised paper (tracked changes).

Specific Comments:

1. Line no. 61-64: Authors can be mentioned about the limitation of the polar orbiting or lower earth orbiting satellites with relevant references. For example, due to poor spatial and temporal resolution of such satellites, there are about 50% data lost over high-altitude sites mountainous sites in particular at 0.05 x 0.05 degree spatial resolution of MODIS (Terra) data (e.g. Ningombam et al., 2021).

**Response:** Thanks for your constructive suggestion. Yes, the orbiting satellites monitoring has many merits, such as the wide spatial coverage. But the low spatial and temporal resolution could not meet the advanced requirements for aerosol information over specific regions. We all agree that the data missing is a big challenge for satellites as the reviewer have mentioned. So we conduct this evaluation work to examine the performance of a multiwavelength photometer, which could be a contributor in ground-based observation in future. According to reviewer's suggestion, the relevant statements have been revised as below.

*Line 63 in revised paper (Line 63 in clean version)*
*"In addition, owing to the limitation of the temporal resolution of satellite-borne platforms over a specific region, such as high-altitude areas and huge-emission areas, satellite AOD retrieval products cannot meet the advanced requirements for ecological environment assessment, heath effect study and real time monitoring."*

*Reference*
*Ningombam, S. S., Song, H. J., Mugil, S. K., Dumka, U. C., Larson, E. J. L., Kumar, B. and Sagar, R.: Evaluation of fractional clear sky over potential astronomical sites, Mon. Not. R. Astron.*

*Soc., 507(3), 3745–3760, doi:10.1093/MNRAS/STAB1971, 2021.*

2. Line no. 79-80: It is also very important to expand such robotic measurement made at high-altitude and mountainous region where there are limited ground-based data available due to harsh climatic condition and lack of manpower support for operating the instruments.

**Response:** The reviewer's suggestion is really appreciated. We have revised corresponding statements as below.

*Line 78 in revised paper (Line 77 in clean version)*
*"These networks have an important role in determining the climatic and environmental effects of aerosols, especially in polar and plateau regions, where the robotic measurements could be a better choice due to the harsh climatic condition and lack of manpower support, and the measurement results have been strictly verified under a wide range of conditions (Dubovik et al., 2000; Eck et al., 1999; Xing et al., 2021a; Zhuang et al., 2017)."*

3. Line no. 134: Please put the unit of water vapor (mm or cm?) after +/- 0.10. Also, I found several places in the manuscript where the authors did not put the unit.

**Response:** Thanks a lot for pointing out. We check through this paper and added the missing unit in corresponding sentences as below.

*Line 146 in revised paper (Line 145 in clean version)*
*"...at 936 nm for water vapor (WV), with uncertainties within ±0.02 and ±0.10 cm, respectively."*

*Line 555 in revised paper (Line 555 in clean version)*
*"Figure 12. The same as Figure 7 and Figure 8 but for water vapor (unit: cm)."*

*Line 592 in revised paper (Line 591 in clean version)*
*"The biases mostly varied within ±0.04 cm, whereas its mean values were concentrated within ±0.02 cm."*

4. Line no. 249: Please mention which version of AERONET data is used as a reference in the present work.

**Response:** Thanks. We used AERONET's results in Version 3.0 as reference in this campaign. We have added this in our revised paper as below.

*Line 274 in revised paper (Line 273 in clean version)*
*"...the cloud-screening results of AERONET as a reference (Version 3.0) ..."*

5. Line no. 306-308: Authors may be added few more relevant references about the importance of quality-controlled data over high-altitude and clean environments where the estimated aerosol

parameters are of the order of measurement uncertainties.

**Response:** The reviewer's comment is really appreciated. We have added some new references here to highlight the importance of data evaluation as below.

*Line 326 in revised paper (Line 325 in clean version)*
*"In terms of AOD evaluation, the key point is that the performance under quite low aerosol loading is largely affected by the instrument accuracy and stability (Campanelli et al., 2007; Che et al., 2009; Kazadzis et al., 2018; Ningombam et al., 2019; Tao et al., 2014)."*

*References*
*Kazadzis, S., Kouremeti, N., Nyeki, S., Gröbner, J. and Wehrli, C.: The World Optical Depth Research and Calibration Center (WORCC) quality assurance and quality control of GAW-PFR AOD measurements, Geosci. Instrumentation, Methods Data Syst., 7(1), 39–53, doi:10.5194/GI-7-39-2018, 2018.*
*Ningombam, S. S., Larson, E. J. L., Dumka, U. C., Estellés, V., Campanelli, M. and Steve, C.: Long-term (1995–2018) aerosol optical depth derived using ground based AERONET and SKYNET measurements from aerosol aged-background sites, Atmos. Pollut. Res., 10(2), 608–620, doi:10.1016/J.APR.2018.10.008, 2019.*

6. Line no. 315: Table 4: PM10 for Level I on 7 November is found to be high. Please check if there are any issues in the data. Moreover, aerosol measurement on the same day for Figure 6 might have disrupted due to frequently passing cloudy which may be attributed the high AOD.

**Response:** The reviewer's suggestion is really appreciated. We checked the data and found they are okay. In fact, we did find there are some days when the $PM_{2.5}$ are low while the high values for $PM_{10}$. However, considering the significant health effects of fine particles, we just used daily $PM_{2.5}$ to defined the air quality in this study. Also, we revised some statements to avoid misleading as below.

*Line 308 in revised paper (Line 307 in clean version)*
*"the daily average $PM_{2.5}$ and $PM_{10}$ concentrations were calculated for the air quality classification with the reference of the ambient air quality standards of China…"*

*Line 311 in revised paper (Line 310 in clean version)*
*"In this study, Level I air quality is defined as…"*

7. Line no. 346: Please correct the wavelength range '70 nm', I think it must be 870 nm.
**Response:** Thanks a lot for pointing out. We have revised this error as below.

*Line 364 in revised paper (Line 365 in clean version)*
*"…at the longer wavelengths of 870, 1020 and 1640 nm."*

8. Figure 12: Please put the unit of water vapor (mm or cm ?) in the Figure. Also, I found several places where the authors did not put the unit.

**Response:** The reviewer's suggestion is really appreciated. We have added the unit for this figure as below.

*Line 146 in revised paper (Line 145 in clean version)*
*"...at 936 nm for water vapor (WV), with uncertainties within ±0.02 and ±0.10 cm, respectively."*

*Line 555 in revised paper (Line 555 in clean version)*
*"Figure 12. The same as Figure 7 and Figure 8 but for water vapor (unit: cm)."*

*Line 592 in revised paper (Line 591 in clean version)*
*"The biases mostly varied within ±0.04 cm, whereas its mean values were concentrated within ±0.02 cm."*

References:
Shantikumar S Ningombam, H-J Song, S K Mugil, Umesh Chandra Dumka, E J L Larson, Brijesh Kumar, Ram Sagar, Evaluation of fractional clear sky over potential astronomical sites, Monthly Notices of the Royal Astronomical Society, 2021, Volume 507(3),pp.3745–3760, https://doi.org/10.1093/mnras/stab1971.

**Response:** The reviewer's suggestion is really appreciated. We have studied this paper and agree that the spatial and temporal resolution is a challenge for satellite monitoring, which the degree of influence is varied from regions to regions. This reference is very useful to highlight the importance of ground-based observation, so we cited this in our revised paper as below.